# Relative humidity over ice as a key variable for Northern Hemisphere midlatitude tropopause inversion layers

**Daniel Köhler**[1,a]**, Philipp Reutter**[1]**, and Peter Spichtinger**[1]

[1]Institute for Atmospheric Physics, Johannes Gutenberg University Mainz, Mainz, Germany
[a]now at: Institute for Atmospheric and Earth System Research, University of Helsinki, Helsinki, Finland

**Correspondence:** Philipp Reutter (preutter@uni-mainz.de)

**Abstract.** The tropopause inversion layer (TIL) is a prominent feature of the midlatitude tropopause region, constituting a transport barrier. Adiabatic and diabatic processes might contribute to the formation and sharpening of the inversion. For both types of processes, relative humidity over ice is ideal for attribution; from theory and former model case studies, we expect enhanced relative humidity values with a sharp TIL.

We use high-resolution radiosonde and ERA5 reanalysis data to show very good qualitative and quantitative agreement in terms of TIL features; thus, coarser ERA5 data can be used for further investigations. Next, we investigate the connection between TIL features and relative humidity measures in both radiosonde and ERA5 data, revealing a clear relationship. Moister profiles, on average, exhibit significantly higher maximum values of the Brunt–Väisälä frequency ($N^2$), indicating a more stable stratification of the tropopause in these cases. This result holds true in both radiosonde measurements and ERA5 data. For TIL thickness, an inverse pattern emerges: moister, more stable TILs exhibit lower thickness.

Because of the good agreement between radiosonde and ERA5 data, we use ERA5 data for seasonal and regional investigations. These analyses reveal consistent TIL properties in various midlatitude regions of the Northern Hemisphere under different meteorological conditions. However, differences in the strength of the dependence of TIL properties on relative humidity over ice are evident between the different regions.

## 1 Introduction

The Earth's atmosphere, a dynamic and complex multiscale system, plays a pivotal role in regulating our planet's climate and weather patterns. Within this intricate atmospheric structure lies the troposphere, the layer closest to the Earth's surface and the site of most of our planet's weather phenomena. At its upper boundary to the adjacent stratosphere, the static stability greatly increases until reaching stratospheric values. This transition layer of increasing stability is often called tropopause region, with more or less strict definitions (see, e.g., Gettelman et al., 2011). About 20 years ago, Birner et al. (2002) were able to demonstrate with high-resolution radiosonde data that sometimes the upper troposphere and lower stratosphere (UTLS) region encounters a strong temperature inversion known as the tropopause inversion layer (TIL). This feature is most prominent if the vertical coordinate system is transformed into a system utilizing the thermal tropopause as the reference point for the vertical coordinate instead of sea level. Sometimes, averaging methods are additionally used to demonstrate the main features on a larger vertical scale.

Situated at the interface between the troposphere and the stratosphere, the TIL represents a unique and enigmatic region characterized by an abrupt increase in temperature with altitude – a significant departure from the typical decrease in temperature observed throughout the troposphere. Thus, a sharp TIL constitutes a strong transport barrier for trace gases, cloud particles and other key variables like vertical motion.

Since its discovery, a couple of hypotheses were developed to explain the origin and formation of the TIL. Wirth and Szabo (2007) showed in model analyses that baroclinic

waves lead to a net sharpening of the tropopause, which leads to a stronger TIL. Gettelman and Wang (2015) and Randel et al. (2007) provided further evidence to support the impact of baroclinic waves on the TIL. Additionally, Randel et al. (2007) suggested a radiative forcing mechanism, where the interaction of ozone and water vapor with radiation contributes to the TIL formation and persistence.

In a recent model study, Kunkel et al. (2016) were able to show that the formation of the TIL is probably driven by a combination of different adiabatic and diabatic processes. In a first step, evolving baroclinic instabilities lead to a compression of isentropes, which in turn results in sharper gradients of the stability. The relevant processes are horizontal convergence in anticyclonic regions; strong upward motions, e.g., triggered by convective instabilities; and gravity waves triggered by the large-scale flow. Since these changes are mostly adiabatic and thus reversible in general, in a second step diabatic processes such as turbulence and/or mixing, cloud formation, and resulting latent heat release and radiative heating or cooling by trace gases (as water vapor) modify the TIL irreversibly. As shown by Kunkel et al. (2016), the diagnostics of these processes are quite difficult, and from measurements it might be quite impossible to disentangle the contributions of the different processes.

However, by carefully inspecting the scenario, there is a quantity which might be considered a proxy for these different processes. The relative humidity (with respect to a stable phase of water, i.e., liquid or solid) is the control variable for many cloud processes. On the other hand, this variable, as combined with the mass concentration of water vapor, pressure and temperature, is a good indicator for adiabatic expansion processes (i.e., cooling); high values of RH can be expected if moist air is adiabatically lifted. In the tropopause region within the low temperature regime (i.e., $T < 235\,\mathrm{K}$), relative humidity over ice (RHi) is the relevant quantity, since solid ice is the stable phase there. Thus, RHi might be a good indicator for the strong lifting of air masses in baroclinic instabilities, thus working as a proxy for strong TILs.

Water vapor is a strong greenhouse gas, especially in the infrared range; the absolute concentration of water molecules controls the amount of emissions and absorption. Particularly in case of a moist layer, we would expect a strong emission of energy in the infrared spectrum and thus a cooling of the layer. However, the total amount of water molecules is not the only reason for strong emissions. Since the atmosphere is layered, the concentration of water vapor in adjacent layers is also of importance. If the layers of different temperatures have a similar amount of water vapor, the emitted radiation is easily absorbed by the layers on top. Thus, a strong gradient of concentration (e.g., a layer with low concentration on top of a layer with high concentration) leads to a much stronger cooling rate than in a situation with weak gradients. Since it is difficult to measure (or determine) the gradient of water vapor concentrations, a good compromise is the use of relative humidity. Since it is linear in the water vapor concentration, it represents the gradients in a meaningful way. Because of small temperature changes in adjacent layers with strong vapor gradients, the impact of the temperature is quite negligible. For the tropopause region, this phenomenon was investigated in a study by Fusina and Spichtinger (2010); a stronger gradient of RHi leads to a much more pronounced cooling on top of the moist layer.

In summary, the use of relative humidity over ice in the tropopause region might help to detect strong TILs. Or in other words, correlations between high values of RHi and strong TILs would corroborate the two-step formation of TILs with adiabatic and diabatic components. Therefore, the use of RHi is highly relevant for the investigation of the tropopause inversion layer. However, a clear distinction between the different processes or the involved timescales is not possible on the basis of RHi values only.

The aim of this study is twofold: first, we want to demonstrate that even with the coarse resolution of ERA5 data it is possible to represent features in the tropopause region, such as the tropopause inversion layer (TIL), in a qualitative way. In addition, the quantitative analysis shows that the absolute values of the TIL properties, e.g., maximum values of static stability, quite closely agree with the values as obtained from high-resolution data (i.e., radiosondes). Second, we want to investigate the correlation of the quantity of relative humidity with the strength of the TIL. Because of the general formation mechanisms of the TIL in terms of adiabatic (dynamic) and diabatic processes, the quantity of relative humidity is the relevant variable, which is related to the formation processes of the TIL. In this study, we make the first attempt to evaluate reanalysis data on a statistical basis, generalizing the findings from case studies and idealized simulations as carried out by Kunkel et al. (2016).

For the first investigation, nearly 10 000 high-resolution radiosonde ascents from one distinct weather station in Germany (Idar-Oberstein) are analyzed. Additionally, this investigation is combined with the ERA5 reanalysis data from the European Centre for Medium-Range Weather Forecasts (ECMWF) for the same location. This approach allows for an evaluation of the quality of ERA5 data concerning the TIL at the presented location. Given that the TIL is associated with strong gradients of stability, the comparison of model data with high-resolution measurements is indispensable.

In the second step, we investigate the correlation between relative humidity and static stability from the ERA5 data in order to corroborate the findings of Kunkel et al. (2016) in a statistical way. Upon successful assessment of data quality, we can extend our analysis to examine the TIL in a similar manner at other locations. We have focused on regions at a similar geographical latitude but with varying frequencies of baroclinic activity. This approach enables us not only to unravel seasonal differences but also to incorporate the influence of atmospheric or even regional peculiarities into the interpretation of the results.

This study is organized as follows. In Sect. 2 we present details on the data and methods used to identify the important quantities of TIL characteristics. Section 3 presents the results. First a comparison of measurements and reanalysis data is provided, followed by an investigation of the influence of the relative humidity on TIL properties. Finally, we extend the examination to geographical and seasonal variations. Conclusions are found in Sect. 4.

## 2 Data and methods

In this section we describe the data sets, the relevant variables and the methods for the statistical investigations.

### 2.1 Data

This study is partly based on radiosonde data from a single measurement site at Idar-Oberstein, Germany (49.69° N, 7.33° E). This site was selected because the German weather service (Deutscher Wetterdienst, DWD) provides 9 years of high-resolution radiosonde data as open access. The exact time frame used spans from 1 January 2011 to 31 December 2019.

The radiosonde measurements are compared with the reanalysis data set ERA5 (Hersbach et al., 2020) provided by the ECMWF. After the comparison and evaluation of the data at the selected site, profiles at different geographical locations are investigated based on the ERA5 data set.

### 2.1.1 Radiosonde data

Idar-Oberstein is 1 out of 12 stations in Germany where the DWD executes synoptic (daily at 00:00, 06:00, 12:00 and 18:00 UTC) high-resolution radiosonde soundings. The station is located at 49.69° N, 7.33° E, and 376 m altitude above sea level. For the radiosonde measurements the Vaisala RS92-SGP (1 January 2011–12 March 2017 and 15 June 2017–31 December 2019) sonde and the Vaisala RS41-SGP (28 March 2017–14 June 2017) sonde are used. The characteristics of the two types of radiosondes are very similar; however, the RS41-SGP has slightly higher precision than the RS92-SGB (https://www.vaisala.com/sites/default/files/documents/RS-Comparison-White-Paper-B211317EN.pdf, last access: 5 September 2024).[TS1] Therefore, the data are treated as if the entire data set is measured by the RS92-SGP radiosonde.

During one ascent of the radiosonde, the meteorological variables are measured with a time resolution of 0.5 Hz[TS2], providing the longitude and latitude with a GPS sensor, the geopotential height ($\Phi_g$, m), the ambient pressure ($p$, hPa), the temperature ($T$, K) and the relative humidity over liquid water (RH, %). In a first approximation, the geopotential height ($\Phi_g$) is equal to the vertical height ($z$). For the considered data set, this approximation is quite good because of Idar-Oberstein's latitude of 49.69° N and our focus on investigations in the UTLS.

This investigation focuses on the upper troposphere and lower stratosphere (UTLS). For obtaining a complete and consistent data set, profiles with a maximum height lower than 20 km and profiles containing missing data are discarded. Over the period from 1 January 2011 to 31 December 2019, the data set contains 10 224 single profiles. A total of 419 profiles are discarded: 311 due to insufficient maximum height, 19 due to missing data, and 89 due to unreliable values of temperature and relative humidity.

The uncertainties of the RS92-SGP regarding the measurements are given by the manufacturer (https://www.bodc.ac.uk/data/documents/nodb/pdf/RS92SGP-Datasheet-B210358EN-F-LOW.pdf, last access: 5 September 2024).[TS3] The temperature sensor has a reaction time less than 2.5 s and a total uncertainty of 0.5 °C. The humidity sensor has a response time between 0.5 and 20 s with a total uncertainty of RH = 5 %. The pressure sensor has a total uncertainty of 1 hPa for 1080 to 100 hPa and 0.6 hPa for 100 to 3 hPa.

The radiosonde humidity data are time-lag-corrected according to Miloshevich et al. (2004), and the water vapor measurements are corrected using the algorithm and coefficients used by Miloshevich et al. (2009). Although the algorithm was developed for the RS92 sonde, it can be applied to the few data points as obtained from the RS41-SGP sonde.

### 2.1.2 ERA5

ERA5 is the most recent reanalysis product of the ECMWF (Hersbach et al., 2020). The reanalysis is a mix of a recalculation of past weather with one fixed forecast model version (IFS CY41R2) and assimilated measurements made for each available time. The high-resolution data set has a horizontal resolution 0.25° in longitude and latitude. The vertical dimension of the atmosphere is represented by hybrid sigma[CE2] (model) levels in ERA5 (Hersbach et al., 2020);[TS4] the number of levels is 137, of which only levels up to the lower stratosphere are used. In the tropopause region, the vertical resolution is about 300 m.

For the comparison with the radiosonde data, we obtained pseudo-radiosonde profiles, i.e., a vertical column at a fixed grid point. The vertical profile is extracted at the 49.75° N, 7.25° E, grid point, which is the closest grid point of ERA5 to the actual location of Idar-Oberstein (49.69° N, 7.33° E). The date and the time of the extracted columns are matched with the reduced radiosonde data set to obtain the maximum comparability between the data sets. The relevant variables, e.g., the geopotential height ($\Phi_g$), are calculated for comparison.

The radiosonde data as described above (Sect. 2.1.1) are also assimilated into the ERA5 data set.

### 2.1.3  Data gridding

In order to guarantee comparability between the radiosonde data and ERA5, it is mandatory to grid them vertically. A regular grid leads to even distances between the data points, which in turn allows for a straightforward statistical analysis. The base of the regular grid is the geometric height ($z$).

Radiosondes use the buoyancy force to ascend; thus, the vertical speed and consequently the vertical resolution are not constant. The buoyancy speed of the used radiosondes ranges from 2 to $8\,\mathrm{m\,s^{-1}}$ (with a mean of around $5\,\mathrm{m\,s^{-1}}$), returning a vertical resolution of 4 to 16 m, respectively, which is usually recognized as high-resolution data (Xu et al., 2023). The final data grid has a 30 m resolution, starting from the station height of 376 m above sea level up to 20 km, in order to reduce the amount of unused data. The interpolation is performed with a cubic spline, which offers sufficient accuracy for this study.

By converting the ERA5 data from a pressure grid to a grid with the geometric height ($z$), the latter grid changes from one point in time to the next with each atmospheric state. Thus, the ERA5 data set is interpolated on the same grid as the radiosondes data (376 m to 20 km, with 30 m resolution) using a cubic spline. The ERA5 data set is heavily over-sampled with a 30 m resolution, meaning the high resolution does not provide additional information; however, the choice is made in order to make the ERA5 data set comparable to the radiosonde data. Finally, we obtained comparable data sets.

## 2.2  Relevant variables

Since most of the desired variables are not directly available, they are calculated from the available variables in the data set. Therefore, the calculation of the relative humidity with respect to ice (RHi, %), the potential temperature ($\theta$, K) and the Brunt–Väisälä frequency ($N^2$, $\mathrm{s^{-2}}$) are described below.

The relative humidity is defined as the ratio of the partial pressure of water vapor ($p_\mathrm{v}$) over the saturation pressure ($p_\mathrm{s}$), which depends on the relevant stable phase (liquid or solid water phase). In this study, the parameterization described by Sonntag (1990) is used for the saturation pressure with respect to liquid water ($p_\mathrm{s,liq}$) and ice ($p_\mathrm{s,ice}$). This choice is motivated by the fact that these formulae are used for radiosonde evaluations by default. The use of RHi as a variable is related to the low-temperature regime in the UTLS (for $T < 240$ K, see, e.g., Reutter et al., 2020) with hexagonal ice as a stable phase of water.

The quantity RHi is derived from RH (relative humidity over liquid water) of the radiosonde using the following relationship:

$$\mathrm{RHi} = \mathrm{RH} \cdot \frac{p_\mathrm{s,liq}(T)}{p_\mathrm{s,ice}(T)}. \tag{1}$$

Note that the most accurate and physically sound formulations for the saturation pressure (over ice or liquid) according to Murphy and Koop (2005) deviate only slightly from the formulae above in the respective temperature regime, leading to a positive bias in the resulting relative humidity over ice of a few percent, which increases with decreasing temperatures. However, for temperatures at the midlatitude tropopause region, this deviation does not crucially affect the investigations.

The variable RHi not only constitutes a measure for atmospheric humidity but also serves as a good proxy for determining the relevant processes, which lead to the formation and further strengthening of the TIL: adiabatic cooling leads to higher values of RHi, and diabatic processes such as cloud formation and radiative feedbacks are also controlled by this quantity. In addition, this humidity variable is a linear variable (in the range between 0 % and about 170 %), which makes the evaluations simpler and more robust than using the specific humidity, which in turn does not allow a relation to cloud processes without additional variables.

The ERA5 data set provides the humidity as the specific humidity ($q$, $\mathrm{kg\,kg^{-1}}$), which is converted to the relative humidity over ice using the following approximation:

$$\mathrm{RHi} \approx \frac{q \cdot p}{\epsilon \cdot p_\mathrm{s,ice}(T)}, \tag{2}$$

with the ratio of the molar masses of water and air $\epsilon = \frac{M_\mathrm{mol,water}}{M_\mathrm{mol,air}} \approx 0.622$.

The potential temperature ($\theta$) is equivalent to the specific entropy of dry air, assuming the ideal gas approximation for dry air. It allows us to compare parcels of air at different pressures levels, and, by definition, it is a conserved quantity for isentropic (i.e., adiabatic) processes. We use the common definition of $\theta$ as stated in Eq. (3) with a constant specific heat capacity $c_p = 1.005\,\mathrm{kJ\,kg^{-1}\,K^{-1}}$ of dry air:

$$\theta = T\left(\frac{p_0}{p}\right)^\kappa \text{ with } \kappa = \frac{R}{c_p} \approx \frac{2}{7}. \tag{3}$$

Here, $p_0 = 1000$ hPa denotes the reference pressure level, and $R = 287.05\,\mathrm{J\,kg^{-1}\,K^{-1}}$ is the specific gas constant for air. This definition with constant $c_p$ is accurate enough for investigations in the tropopause region (see discussion in Baumgartner et al., 2020). The relation to specific entropy of dry air is given by $\mathrm{d}s = c_p\mathrm{d}\log(\theta)$.

The static stability of dry air or, more commonly, the Brunt–Väisälä frequency squared ($N^2$) is a common measure for the stability of the dry atmosphere (Hantel, 2013). $N^2 < 0$ characterizes an unstable stratification, $N^2 = 0$ characterizes a neutral stratification and $N^2 > 0$ characterizes a stable stratification. The free troposphere is dominantly stable, and the stratosphere is considerably more stable than the troposphere. The static stability is derived from the buoyancy force (i.e., Archimedes' principle) and can be approximate by

$$N^2 = \frac{g}{\theta} \cdot \frac{\partial \theta}{\partial z} = \frac{g}{T}(\Gamma_\mathrm{d} - \Gamma), \tag{4}$$

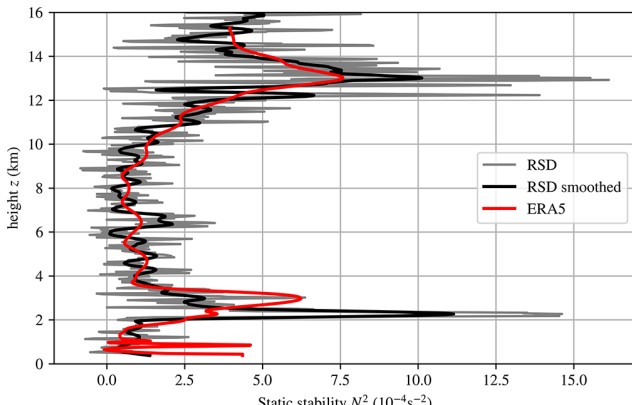

**Figure 1.** Example of a static stability profile from 2 January 2011, 00:00 UTC, with (gray) the calculated $N^2$ from radiosonde data, (black) the 330 m window smoothed $N^2$ from radiosonde data and (red) the calculated $N^2$ from ERA5 data.

with $g = 9.8066\,\mathrm{m\,s^{-2}}$ the (mean) gravitational acceleration, $\Gamma_d = \frac{g}{c_p}$ the dry adiabatic lapse rate and $\Gamma = \frac{\partial T}{\partial z}$ the actual temperature lapse rate based on the geometric height ($z$).

This approximation works under the assumption of dry air and returns on average values that are too high for the Brunt–Väisälä frequency; moisture leads to a strong decrease in the static stability, even if no phase change is triggered (Durran and Klemp, 1982). However, a moist and commonly accepted analog to dry static stability is still missing, although there have been some attempts at a consistent treatment (Peters et al., 2022). Therefore, we use dry static stability to ensure comparable results with the literature that use the dry approximation (e.g., Gettelman and Wang, 2015; Birner et al., 2002; Birner, 2006; Erler and Wirth, 2011).

As the measurements of a radiosonde are discrete, a numerical approximation of the derivative is necessary. Since the grid increments are quite small, the numerically derived gradients are highly variable. Thus, a centered approximation of the fourth order,

$$\frac{\partial \theta}{\partial z} \approx \frac{4}{3}\frac{\theta_{z+1} - \theta_{z-1}}{z_{z+1} - z_{z-1}} - \frac{1}{3}\frac{\theta_{z+2} - \theta_{z-2}}{z_{z+2} - z_{z-2}}, \tag{5}$$

is used in order to smooth the resulting finite gradient approximations. However, even this high-order method leads to a highly variable $N^2$ profile. For a better handling, the profile is additionally smoothed using a running mean with a window of 330 m, as can be seen in Fig. 1.

## 2.3 Calculation of tropopause characteristics

### 2.3.1 Definition of tropopause

The tropopause separates the troposphere and the stratosphere, constituting a transport barrier for trace gases and cloud particles. There have been several attempts to define the tropopause using different ways. We list the most com-

mon approaches; however, afterwards we will use the classical definition of a thermal tropopause as defined by the World Meteorological Organization (WMO) in the middle of the last century. For more details about the history and difficult definition of the tropopause, we refer to the studies by Hoinka (1997), Maddox and Mullendore (2018), Tinney et al. (2022) and Hoffmann and Spang (2022).

The tropopause can be defined on the basis of different variables: physical temperature, potential temperature, potential vorticity (PV) or even chemical trace gases such as ozone. In the classical definition by the WMO (1957), the lapse rate of the temperature is taken into account. The full criterion can be stated as follows: CE3

> The *first tropopause* is defined as the lowest level at which the lapse rate decreases to 2 °C/km or less, provided also the average lapse rate between this level and all higher levels within 2 km does not exceed 2 °C/km. (WMO, 1957)

For the tropics, sometimes the so-called cold point temperature is used, i.e., the first minimum in the free troposphere of the physical temperature (Highwood and Hoskins, 1998).

In a further step, the potential temperature ($\theta$) can be used as a basis variable. The gradient of $\theta$ can be used as a simple measure for the static stability, thus influencing tracer transport (see, e.g., Kunz et al., 2011). The tropopause can be defined as the height level at which a certain threshold in the gradient $\frac{\partial \theta}{\partial z}$ exceeds a certain threshold (e.g., $\frac{\partial \theta}{\partial z}\big|_{\mathrm{thres}} \sim 0.012\,\mathrm{K\,m^{-1}}$; see Mullendore et al., 2005). Following the discussion by Tinney et al. (2022), it seems that a refined version of this definition might be most robust. The Brunt–Väisälä (or buoyancy) frequency can be calculated from the potential temperature; see Eq. (4). Instead of using the Brunt–Väisälä frequency itself, Birner (2010) introduced the level of the maximum gradient of $N^2$ as the tropopause; actually, this is the level of the maximum curvature of the temperature, which can be set in relation to the residual circulation.

From an atmospheric dynamics perspective, the potential vorticity as an adiabatic invariant is investigated. The level of a certain (but kind of arbitrary) threshold is the set as the dynamical tropopause. While often the threshold of 2 PVU is used, Kunz et al. (2011) showed that the value in terms of transport barriers can vary between 1.5 and 5 PVU depending on the season. The threshold of 3.5 PVU often produces a dynamical tropopause height which is close to the thermal tropopause level as derived by the WMO criterion (Hoerling et al., 1991).

Finally, chemical trace gases are used to define the tropopause as a transport layer. Actually, from this point of view the tropopause is not an ideal clearly defined interface between troposphere and stratosphere but merely a transition layer (see, e.g., the discussion in Gettelman et al., 2011). The depth of the transition layer was investigated using tracer–tracer correlation, e.g., using ozone and carbon monoxide (Pan et al., 2004). Since there is a clear signal in the ozone

concentration in the different vertical layers (troposphere vs. stratosphere), a threshold criterion might be used to define the ozonopause, which should be close to tropopause levels derived from other definitions. Bethan et al. (1996) used ⁵ a threshold of ozone mixing ratio on the order of $\sim 100$–110 ppb and showed that the ozonopause is usually quite close to the thermal tropopause. This approach was used further for a simple discrimination of aircraft data (see, e.g., Gierens et al., 1999) and for other applications without us-¹⁰ ing a full 3D data set of meteorological variables. In some investigations, the hygropause (i.e., the minimum in water vapor concentrations) was also used as a proxy. However, this approach is not really successfully applied for data analysis. Although there might be more modern definitions of the ¹⁵ tropopause level, we stick with the classical definition by the WMO (1957). This is mostly due to a better comparison with former studies on the tropopause inversion layer (e.g., Birner et al., 2002; Birner, 2006); however, this definition represents the nature of the transport barrier very well and is still a stan-²⁰ dard definition for the daily reports of all weather services in the world.

One should keep in mind that almost all definitions of the tropopause level were driven by the large-scale perspective of atmospheric dynamics, i.e., using the viewpoint that the ²⁵ vertical change in the thermodynamic variables (i.e., temperature and pressure) is smooth enough. This viewpoint is clearly represented in the WMO definition using lapse rates, where changes over a long vertical extent are investigated. In general, this definition requires a certain vertical resolution, ³⁰ whereas deviations in both directions might raise issues. For very coarse resolution data as in low-resolution radiosonde reports, former reanalysis data or climate models, the determination of lapse rates causes problems, which can be handled with some refined methods (see, e.g., Hoinka, 1998; Re-³⁵ ichler et al., 2003).

For high-resolution data, as in operational radiosondes or partly in the new generation of reanalysis data (see, e.g., a similar discussion about front detection in Niebler et al., 2022), the variables are not smooth enough for the deter-⁴⁰ mination of gradients; actually, the high resolution leads to strong variations or even nonphysical noise. In extreme cases, the WMO criterion is never fulfilled, since the lapse rate crucially changes within the required extension of 2 km; see also the discussion in Maddox and Mullendore (2018). ⁴⁵ For high-resolution data, averaging (e.g., running means) of high-resolution data or higher-order finite-difference methods for determining the gradients can be applied; see, e.g., the calculation of $\frac{\partial \theta}{\partial z}$ in Eq. (5). However, the question of whether the definition of a tropopause layer as driven by the ⁵⁰ large-scale viewpoint of atmospheric dynamics is still meaningful if we consider much smaller scales remains. At least for investigations of convective events, some effort is made to find more robust measures for the tropopause characteristics (see, e.g., Tinney et al., 2022).

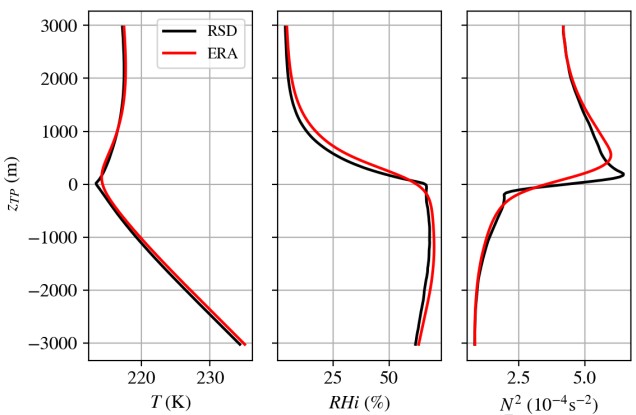

**Figure 2.** The average vertical profile in the tropopause relative height ($z_{\mathrm{TP}}$) coordinate system of temperature ($T$), relative humidity over ice (RHi) and static stability ($N^2$) for the radiosondes (black) and ERA5 (red).

In this paper we use the pure WMO criterion to determine ⁵⁵ the tropopause. For details, the reader is referred to the source code that we have provided (https://doi.org/10.5281/zenodo. 10604349, Köhler, 2024).

### 2.3.2   Relative coordinates

In order to be able to compare the large number of ra-⁶⁰ diosonde data and ERA5 profiles, a tropopause-centered coordinate system was introduced. For this purpose, the thermal tropopause is identified in each radiosonde and ERA5 profile for Idar-Oberstein and defined as the tropopause height (TP$_z$). This means that all profiles are now in the same co-⁶⁵ ordinate system and can be averaged to obtain mean profiles of temperature, humidity and static stability. Therefore, we introduce a new height variable,

$$z_{\mathrm{TP}} = z - \mathrm{TP}_z, \tag{6}$$

relative to the tropopause height (TP$_z$; as derived by the ⁷⁰ WMO criterion; see above). Negative altitude values of $z_{\mathrm{TP}}$ denote the upper troposphere, whereas positive altitude values of $z_{\mathrm{TP}}$ represent the lower stratosphere. For the averaging process the single profiles are transformed into the $z_{\mathrm{TP}}$ coordinate system, and the arithmetic mean of a meteorological ⁷⁵ variable $\chi \in$ {temperature, relative humidity, static stability} is calculated, summing over all profiles at a certain height.

The mean profiles relative to the tropopause height of temperature, static stability ($N^2$) and RHi can be seen in Fig. 2 for the radiosonde measurements (black) and the correspond-⁸⁰ ing ERA5 data set (red) at the location of Idar-Oberstein. Even for the mean profiles, the characteristics of the TIL, i.e., the strong increase in $N^2$ at around TP$_z$, can be seen clearly, as described in the next section. However, Fig. 2 also shows that the results differ between radiosonde and reanalysis data. ⁸⁵ In the temperature profile of ERA5, the minimum temperature is less pronounced, as well as the values in RHi at the

tropopause level. The static stability profile shows a deviation of the maximum in $N^2$; i.e., the level of the maximum is shifted towards higher altitudes, and the maximum is less pronounced as compared to the high-resolution radiosonde
data. These differences are mainly based on the vertical resolution of the data sets, which is significantly lower in the case of the ERA5 data than in the radiosonde data. As a result, sharp gradients cannot be resolved as well, as can be seen in particular when looking at $N^2$. A detailed compari-
son between the radiosonde and ERA5 data can be found in Sect. 3.1.1.

### 2.3.3  Tropopause inversion layer

The tropopause inversion layer (TIL) is a region of extraordinarily high static stability within the tropopause region and
15 is found by averaging vertical profiles with respect to the tropopause level. The TIL is also present in single vertical profiles of radiosondes (Birner et al., 2002) and models (Birner, 2006). The high stability in the TIL region represents a barrier to vertical motion (Gettelman et al., 2011) and
20 is therefore important for understanding the composition of the air in the upper troposphere and lower stratosphere. Although there have been investigations of the TIL properties for over 2 decades, a clear (or even common) definition of the TIL and its main features (e.g., strength) is missing. For this
study, we define the TIL strength (sTIL) as the maximum of the static stability ($N^2_{\max}$) within 3 km above tropopause level and the altitude level of $N^2_{\max}$ as the TIL height (TIL$_z$). Two additional heights are defined through two minimum values of $N^2$: UT-$N^2_{\min}$ is the height of the minimum of $N^2$ in the
upper troposphere and LS-$N^2_{\min}$ is the height of the minimum of $N^2$ within 5 km above TIL height. The TIL depth (dTIL) is half of the height difference between the UT-$N^2_{\min}$ and LS-$N^2_{\min}$. The diagnostics of the TIL with the main features and the newly introduced quantities sTIL and dTIL are summa-
rized in Fig. 3. If one of the features, such as UT-$N^2_{\min}$ or LS-$N^2_{\min}$, could not be determined, these profiles were excluded from consideration in this study. This affected 126 profiles, so that in the end 9678 profiles were included in the analysis in this study. Note that the real profile of high-resolution
data (radiosonde and ERA5) as represented in Fig. 1 includes all the features of the scheme shown in Fig. 3 (UT minimum of $N^2$, maximum of $N^2$, LS minimum of $N^2$). After averaging over many tropopause-centered profiles, some features might be lost in the mean profiles (e.g., Fig. 2), although they
are still visible in the single profiles – otherwise the profiles would be discharged in the analysis.

### 2.3.4  Calculation of humidity measures

The analysis of the humidity of the upper troposphere and the lower stratosphere is based on the average humidity with
50 respect to ice below the TIL, denoted by wRHi. It is calculated by averaging RHi from the height of the 500 hPa pres-

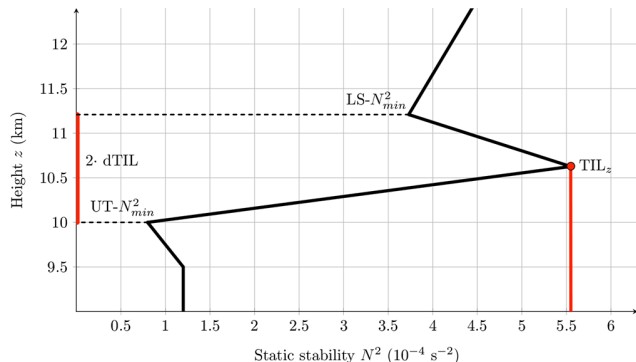

**Figure 3.** A schematic drawing of the different diagnostics of the tropopause inversion layer, i.e., the TIL depth (dTIL) and the strength of the TIL (sTIL). These features (minima and maximum in $N^2$) can also be seen in the realistic profile represented in Fig. 1.

sure surface ($z_{p500}$) up to the TIL level (TIL$_z$), as defined in Eq. (7).

$$
\begin{aligned}
\text{wRHi} &= \frac{1}{\text{TIL}_z - z_{p500}} \int_{z_{p500}}^{\text{TIL}_z} \text{RHi}(z)\mathrm{d}z \\
&\approx \frac{1}{\text{TIL}_z - z_{p500}} \sum_{z=z_{p500}}^{\text{TIL}_z} \text{RHi}(z)\Delta z
\end{aligned}
\tag{7}
$$

This moisture (or humidity) is used to sort the vertical profiles according to different moisture contents. 55

## 3  Results

Based on the spatially and temporally highly resolved ERA5 data, the properties of the tropopause region related to the static stability and relative humidity in this region are investigated in more detail. As stated in the introduction, our goal 60 of this study is twofold. First, we want to show that there is good agreement of high-resolution radiosondes and ERA5 data in terms of representing the main features of TILs in the midlatitudes. Therefore, the measured data of the radiosondes are compared with the corresponding data of the reanal- 65 ysis model. Second, we want to show that relative humidity is a key quantity for TILs; thus, there is a strong correlation between high-humidity measures and sharp and strong TILs. This is investigated in more detail, also with the considera- 70 tion of seasonal and geographical differences.

### 3.1  TIL properties in measurements and reanalysis data

Very good agreement between the radiosonde measurements and the reanalysis data is the basic prerequisite for further investigations based on the ERA5 data. Therefore, in a first 75 step, the deviations between radiosonde measurements and reanalysis data for the variables temperature ($T$), relative humidity with respect to ice (RHi) and static stability ($N^2$) are

investigated. In the next step, the results for the different heights ($TP_z$, $TIL_z$) are investigated. Finally, TIL properties such as TIL thickness and TIL depth are compared.

### 3.1.1 Comparison of temperature and relative humidity with respect to ice

The deviation between radiosonde measurements and ERA5 data in a variable $\chi$ is quantified by the average measure $\overline{D}_{abs}(\chi)$ for every single profile. This quantity can be calculated as follows:

$$\overline{D}_{abs}(\chi) := \frac{1}{z'-z_0} \int_{z_0}^{z'} |E(\chi) - R(\chi)| \mathrm{d}z$$

$$\approx \frac{1}{z'-z_0} \sum_{z_0}^{z'} |E(\chi(z)) - R(\chi(z))| \Delta z, \qquad (8)$$

with $z_0$ the start height and $z'$ the end height of the averaging, $\chi$ the meteorological variable of interest, $E$ the ERA5 profile, and $R$ the radiosonde profile. $\Delta z$ is the height difference between two adjacent levels. We chose a metric including absolute values of differences in order to avoid undesired cancellation effects of positive and negative contributions. In this sense, we used a metric inspired by the $L_1$-norm. Thus, the resulting distributions are expected to be skewed and might have (exponentially) decaying tails. The resulting data set is visualized in a probability bar chart, and the corresponding median, mean and standard deviation for the different variables are presented in Figs. 4, 5 and 6, respectively.

The temperature deviations between the radiosondes and ERA5 for the upper troposphere ($z_0 = -3000\,\mathrm{m}$, $z' = 0\,\mathrm{m}$, Fig. 4a) show a skewed distribution with a median of $0.37\,\mathrm{K}$, a mean of $0.43\,\mathrm{K}$ and a standard deviation $\sigma = 0.22\,\mathrm{K}$. For the lower stratosphere ($z_0 = 0\,\mathrm{m}$, $z' = 3000\,\mathrm{m}$) the distribution of the temperature deviation is similar (Fig. 4b). However, the median ($0.67\,\mathrm{K}$) and mean ($0.73\,\mathrm{K}$) values are significantly larger, with a standard deviation of $\sigma = 0.26\,\mathrm{K}$. Generally we know that ERA5 has on average slightly warmer temperatures compared to the measurements. In order to interpret these results, it should be mentioned that the measurements have an uncertainty of $0.5\,\mathrm{K}$ according to the manufacturer (https://www.bodc.ac.uk/data/documents/nodb/pdf/RS92SGP-Datasheet-B210358EN-F-LOW.pdf). TS5.

For deriving a robust statement about the humidity impact on the TIL in the tropopause region, the relative humidity with respect to ice (RHi) is used. As mentioned above, this is the key thermodynamic control variable for ice cloud processes, thus determining also the life cycle of ice clouds. The consistency of the moisture data is also important for the description of the average relative humidity with respect to ice (wRHi). The distributions in Fig. 5 for mean differences show a shift to higher deviations of RHi in ERA5. This behavior is due to the fact that ERA5 data do not capture the moisture gradients at the tropopause as sharply as the radiosonde data. The humidity features, i.e., the fine structures, are smeared out; this is partly due to the coarse vertical resolution of the ERA5 data but probably also due to issues in the data assimilation of moisture in cold temperature regimes.

This difference dominates the value of $\overline{D}_{abs}(\chi)$ for each individual vertical profile; thus, we would expect less steep gradients for RHi in the ERA5 data. These differences are again distributed with some skewness, leading to a quite similar median value (12.95 %) and mean value (13.99 %) in the upper troposphere and accordingly a similar median value (2.96 %) and mean value (3.56 %) in the lower stratosphere. The standard deviation for the upper troposphere is $\sigma = 6.57\,\%$, which is a considerable deviation, where relative humidity over ice ranges from 0 %–105 %. In the lower stratosphere, where relative humidity is generally much lower than in the troposphere, the standard deviation is $\sigma = 2.49\,\%$.

The static stability or Brunt–Väisälä frequency squared ($N^2$) represents the main criterion to identify the tropopause inversion layer and its characteristics. Looking at the probability distribution (Fig. 6), there exists a notable difference between the upper troposphere and the lower stratosphere. In the upper troposphere (Fig. 6a), ERA5 has a tendency to be more stable with a mean $= 0.40 \times 10^{-4}\,\mathrm{s}^{-2}$ and a median $= 0.43 \times 10^{-4}\,\mathrm{s}^{-2}$. The distribution is skewed towards smaller average difference values of $N^2$ with a standard deviation of $\sigma = 0.15 \times 10^{-4}\,\mathrm{s}^{-2}$. When comparing the average differences and the standard deviation of $N^2$ to the static stability for the upper troposphere $1.2 \times 10^{-4}\,\mathrm{s}^{-2}$ (Hoskins and James, 2014), the static stability is represented well in the ERA5 reanalysis.

In the lower stratosphere (Fig. 6b), ERA5 is less stable compared to the radiosonde data with a mean $= 1.02 \times 10^{-4}\,\mathrm{s}^{-2}$ and a median $= 1.07 \times 10^{-4}\,\mathrm{s}^{-2}$, with a tendency to smaller absolute values of static stability; the standard deviation is $0.31 \times 10^{-4}\,\mathrm{s}^{-2}$.

The reason for the lower stability in the lower stratosphere in ERA5 is the tropopause inversion layer, which arises through a strong gradient of potential temperature. Due to the lower vertical resolution of ERA5, strong gradients of the variables are less pronounced, leading to smaller values of static stability, thus resulting in a less stable vertical profile in the lower stratosphere. Nonetheless, the average difference of $\sim 10^{-4}\,\mathrm{s}^{-2}$ is small compared to the average value of $N^2$ of $4 \times 10^{-4}\,\mathrm{s}^{-2}$.

Overall, we can state that the quantitative agreement between the high-resolution radiosonde data and the ERA5 data is high enough to represent the profiles of $T$, RHi and $N^2$ in a satisfying way. However, the more important issue in the comparison is the qualitative representation of TIL features in both data sets, as will be investigated in the next section.

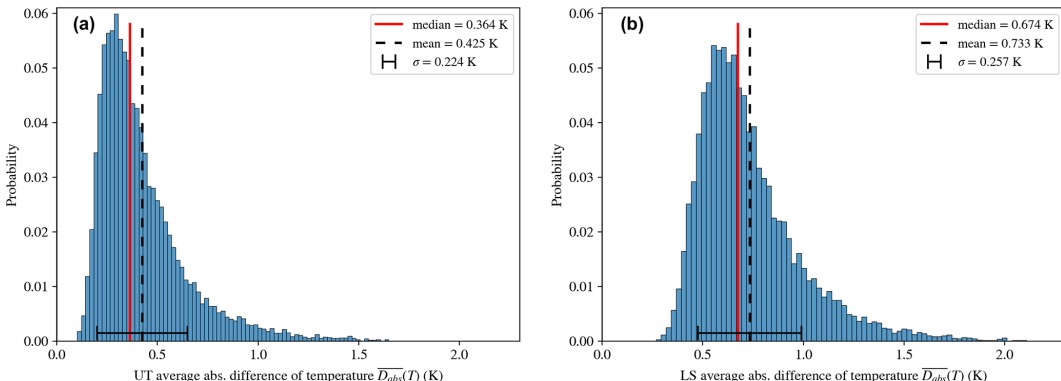

**Figure 4.** Probability distribution of the average difference in temperature ($\overline{D}(T)$) between ERA5 and radiosondes from Idar-Oberstein for the upper troposphere (UT; $z_0 = -3000\,\text{m}$, $z' = 0\,\text{m}$; **a**) and the lower stratosphere (LS; $z_0 = 0\,\text{m}$, $z' = 3000\,\text{m}$; **b**). The median is displayed in red, the mean is represented by a dashed black line and the standard deviation ($\sigma$) is represented as an error bar.

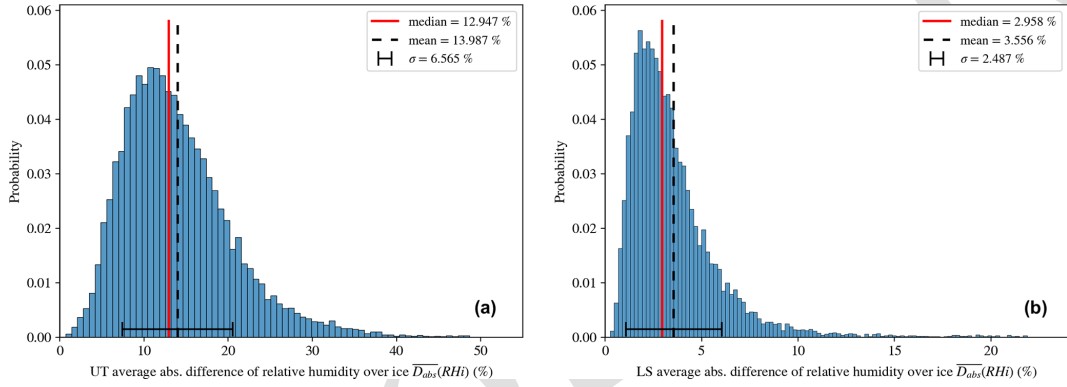

**Figure 5.** Probability distribution of the average difference in relative humidity over ice ($\overline{D}(\text{RHi})$) between ERA5 and radiosondes from Idar-Oberstein for the upper troposphere (UT; **a**) and the lower stratosphere (LS; **b**), with the median in red, the mean in dashed black and the standard deviation ($\sigma$) as an error bar.

## 3.2 TIL properties and humidity

In the following sections we investigate the relationship between TIL properties and moisture, especially in terms of TIL strength and thickness. As a measure for humidity we use the averaged relative humidity with respect to ice (wRHi) as introduced earlier in Eq. (7).

### 3.2.1 TIL strength and humidity

First, we classify the vertical profiles into tropopause inversion layers of different strengths (sTIL). The classification of sTIL is based on three classes, i.e., low, medium and high values of sTIL, with different intervals for the radiosonde data and the ERA5 data. The ranges are represented in Table 1. The classification criteria were chosen such that one-third of the vertical profiles fall into each category (low, medium, high). Since the distributions of metrics between ERA5 and radiosondes differ, the exact values of the classification boundaries are also different. Figure 7 shows the cor-

responding mean profiles for temperature, RHi and static stability for the radiosondes (left) and the reanalysis data (right).

As expected, the ERA5 reanalysis is not able to capture the sharp gradients as well as the high-resolution radiosondes. This is noticeable with the sharp kinks in temperature ($T$) and relative humidity with respect to ice (RHi) at the tropopause level ($z_{\text{TP}} = 0\,\text{m}$) and the sharp spikes in the static stability ($N^2$), which are present in the radiosonde data but are smoothed out in the ERA5 data. Despite these issues related to the resolution of ERA5, the reanalysis data show the same qualitative behavior for temperature and RHi. Thus, the ERA5 data set can be used consistently for the investigation of TIL in the midlatitudes.

With the focus on the temperature profile, stronger TILs seem to be correlated with colder temperatures and thus possibly with higher tropopause heights. However, this interpretation is somewhat problematic, since warmer and colder profiles with higher and lower tropopause heights are compared. Some altitude shifts in the profiles might weaken or even cancel out these effects. The correlation between a

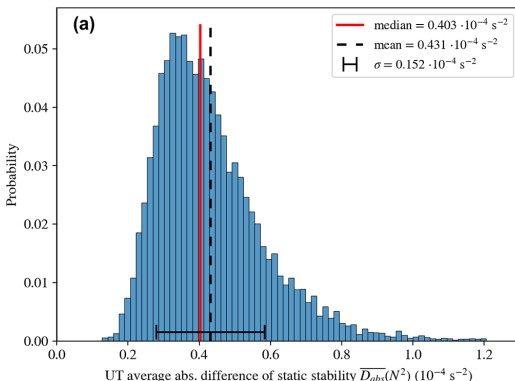 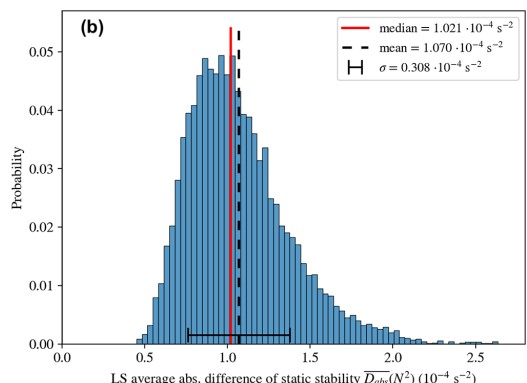

**Figure 6.** Probability distribution of the average difference in static stability ($\overline{D}(N^2)$) between ERA5 and radiosondes from Idar-Oberstein for the upper troposphere (UT; **a**) and the lower stratosphere (LS; **b**), with the median in red, the mean in dashed black and the standard deviation ($\sigma$) as an error bar.

**Table 1.** Values of sTIL for the three different classes (low, medium, high) in the radiosonde (RS) data set and the ERA5 data set.

| sTIL | Low | Medium | High |
|---|---|---|---|
| RS | $< 6.8 \times 10^{-4}\,\mathrm{s}^{-2}$ | $[6.8 \times 10^{-4}\,\mathrm{s}^{-2}, 11.2 \times 10^{-4}\,\mathrm{s}^{-2}]$ | $> 11.2 \times 10^{-4}\,\mathrm{s}^{-2}$ |
| ERA5 | $< 5.2 \times 10^{-4}\,\mathrm{s}^{-2}$ | $[5.2 \times 10^{-4}\,\mathrm{s}^{-2}, 8.4 \times 10^{-4}\,\mathrm{s}^{-2}]$ | $> 8.4 \times 10^{-4}\,\mathrm{s}^{-2}$ |

sharper temperature inversion above the tropopause and a stronger TIL is certainly expected, because the TIL strength is derived directly from the temperature gradients (Eq. 4).

Thus, we shift our investigations to the connection between relative humidity and TIL strength. Here, we find two robust features, which can be clearly seen in the different classes of the TIL strength (sTIL). For the class with the highest sTIL values (green line in Fig. 7), we find enhanced values of RHi throughout the troposphere until the tropopause level; here, the averaged values of RHi are usually above 70 %. For the medium class, the relative humidity values show averaged values of about 60 %, whereas in the low-sTIL class, the relative humidity values are around 50 %. The signal is the same for radiosonde data and ERA5 data, with slight changes in the values.

In addition, we find that for the high-sTIL class the RHi gradient on top of the moist layer is much sharper than for the other two classes. Again, we see a clear decrease in the RHi gradient for the medium- and low-sTIL classes.

In summary, we find a sharper or stronger TIL for

1. higher RHi values in the troposphere and

2. sharper gradients of RHi on top of the moist layer (at the tropopause height and above).

The first feature of enhanced RHi values (with a slight increase with height towards the tropopause level) can be interpreted as a signal of the adiabatic processes (e.g., during baroclinic instability evolution) enforcing a vertical upward motion (with compressing isentropes) and thus an increase in RHi. A higher value of RHi might also indicate a stronger

**Table 2.** Maxima in $N^2$ for the different classes of wRHi as represented in Fig. 8.

| $N^2_{\max}$ | Low wRHi | Medium wRHi | High wRHi |
|---|---|---|---|
| RS | $7.3 \times 10^{-4}\,\mathrm{s}^{-2}$ | $8.2 \times 10^{-4}\,\mathrm{s}^{-2}$ | $9.3 \times 10^{-4}\,\mathrm{s}^{-2}$ |
| ERA5 | $5.7 \times 10^{-4}\,\mathrm{s}^{-2}$ | $6.5 \times 10^{-4}\,\mathrm{s}^{-2}$ | $7.0 \times 10^{-4}\,\mathrm{s}^{-2}$ |

(or even longer) adiabatic process. The second feature of a strong moisture gradient at the tropopause level might point to diabatic processes leading to irreversible formation of the TIL. Actually, Fusina and Spichtinger (2010) could show that the strongest radiative cooling in moist tropopause layers was triggered for strong RHi gradients. Thus, the variable RHi points to both processes according to higher values and steeper gradients.

Another way to investigate the relationship between humidity and TIL strength is to look at the sTIL distribution as a function of the mean RHi. For this purpose, the sTIL data of the radiosondes and of ERA5 were divided into three RHi classes: low wRHi (wRHi $\leq 45$ %), medium wRHi (45 % $<$ wRHi $\leq 70$ %) and high wRHi (70 % $<$ wRHi). These distributions are represented in Fig. 8 for radiosonde data (left panel) and ERA5 data (right panel).

We find in Fig. 8 that the probability density function (PDF) of the TIL strength is correlated with higher values of averaged relative humidity with respect to ice (wRHi). This is deducible in the shift of the PDF curve to higher sTIL values with higher wRHi categories. This shift is most obvious

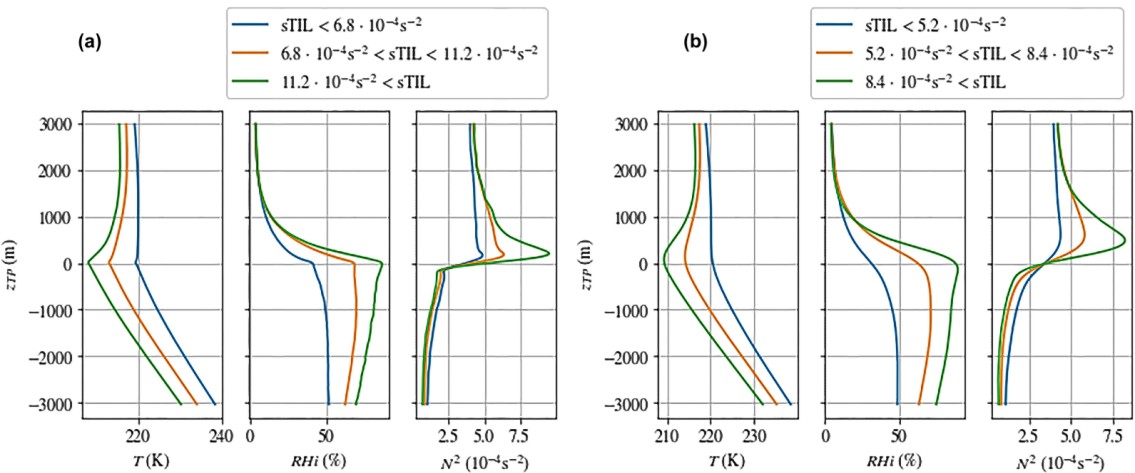

**Figure 7.** The mean vertical profiles for radiosondes **(a)** and ERA5 **(b)** of temperature ($T$), relative humidity over ice (RHi) and static stability ($N^2$) in the tropopause relative height ($z_{TP}$). The mean profiles are classified in terms of tropopause inversion layer strength (sTIL). Colors indicate the different sTIL categories: blue, low-sTIL-value category; orange, medium-sTIL-value category; green, high-sTIL-value category. The exact boundary values of the categories are found in the respective legend.

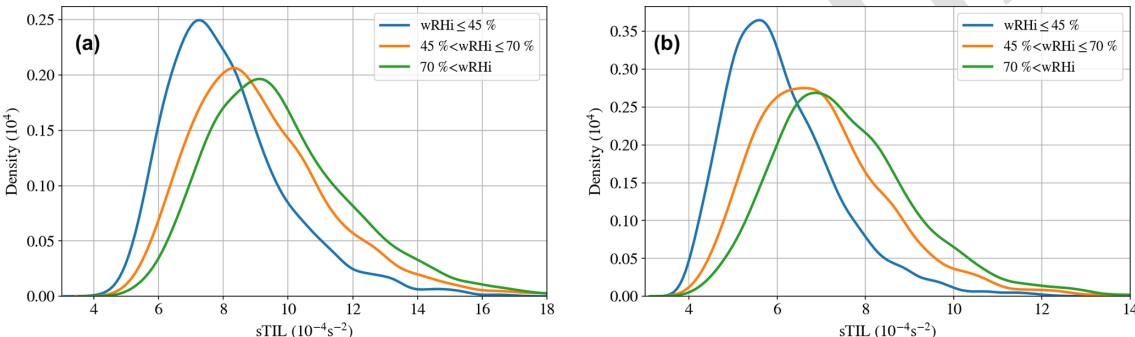

**Figure 8.** The probability density function for the radiosondes **(a)** and ERA5 **(b)** of the TIL strength (sTIL) for categories of the average relative humidity over ice (wRHi). Colors indicate the different wRHi categories: blue, low category (wRHi $\leq 45\,\%$); orange, medium category ($45\,\% <$ wRHi $\leq 70\,\%$); green, high category ($70\,\% <$ wRHi). Note the slightly different scale in the sTIL axis in both panels.

at the position of the maximum of the mode of the PDF, as can be seen in both data sets – the high-resolution radiosonde data (Fig. 8a) and the ERA5 data (Fig. 8b). The values of the maximum of the modes for the different categories are listed in Table 2. It is important to mention that the PDF broadens with higher wRHi, which means that the variance of the sTIL values increases. The increased variance does not lead the different PDF to cross in the left tail of the function, meaning that weak sTIL values always have a higher probability of being in the lower-humidity category. Similarly, higher sTIL values ($> 9.3 \times 10^{-4}\,\mathrm{s}^{-2}$ for radiosondes and $> 6.5 \times 10^{-4}\,\mathrm{s}^{-2}$ for ERA5) have a higher probability of being part of the high-wRHi category.

When comparing the radiosondes (Fig. 8a) and ERA5 (Fig. 8b), the similar shape in the probability density and the same trends are apparent, while the values of sTIL are shifted to lower values when comparing ERA5 with the radiosondes.

Again we find the same qualitative behavior for RS data and reanalysis data, although the quantitative values differ due to the different vertical resolution of the data. Overall, we can state that there is a strong indication that high RHi values are physically connected with strong TILs. This result is in agreement with the findings by Kunkel et al. (2016) for a single case analysis; adiabatic processes (transport by atmospheric flows) and diabatic processes (as mixing or radiative cooling) lead to enhanced values of RHi.

### 3.2.2 TIL depth and humidity

In addition to the strength of the TIL, we show in Fig. 9 the TIL depths (dTIL) distributed among the different classes of averaged relative humidity with respect to ice (wRHi). Here, the depth of the TIL also shows a correlation with the RHi, which is discussed in this section. The total width of the distribution does not change, so the depth of the TIL is always in

the range from some few hundred meters up to about 6000 m; however, the maximum of the distribution shows a clear deviation. As the average relative humidity over ice (wRHi) increases, the TIL depth (dTIL) is shifted to lower values.

The probability density function shows a formation of a second mode, as seen in the ERA5 data set for all wRHi classes (Fig. 9b), as well as in the low-wRHi category of the measurements (blue line, Fig. 9a). This formation of a double mode is an artifact of the categorization process, in which the aim was to ensure that one-third of the data per category were included. These double modes are therefore not of a physical nature. The origin is a fluctuation in the distribution, which is cut off by the boundaries of the categorization. The artifact makes the interpretation of the mode difficult. Nonetheless there is a clear correlation between higher dTIL values and lower wRHi values. The variance of the data within each wRHi category is about the same, as seen by the similar width and height of the PDF.

Despite the different resolutions of the underlying data, the results of the ERA5 and the radiosonde data are very similar. The maximum of the distributions for the high-wRHi category is approximately 2250 m in both cases. In contrast, the thickness of the TIL in the driest category, low wRHi, is around 3000 m. The category in the middle with wRHi between 45 % and 70 % is about 2900 m in both data sets. As mentioned above, the double mode makes exact quantification difficult.

In summary, it can be seen that the drier the air in this region is, the thicker the TIL is. In other words, a moist upper troposphere coincides with a decreased depth of the TIL. In combination with the findings from the section before, we can conclude that for a more humid upper troposphere, we can find a stronger but vertically more confined TIL feature. This confinement might be driven by radiative processes, since a sharp moisture gradient at the tropopause level leads to a strong but vertically very confined radiative effect (see, e.g., Fig. 9 in Fusina and Spichtinger, 2010). In addition, a strong vertical upward motion as triggered by adiabatic processes might also lead to a confinement of the vertical TIL features. However, a clear attribution of this connection is not possible, although the enhanced humidity points again to the adiabatic and diabatic processes. With our Eulerian approach of evaluating RHi at a given location and time, a further determination of the dominant processes and/or their timescales is not possible. For such investigations, a Lagrangian approach would be necessary.

## 3.3  Geographical variations

We found a correlation between TIL characteristics and humidity for the Idar-Oberstein site in central Europe with radiosonde measurements and reanalysis data; thus, the next step is to investigate three additional regions representative of Northern Hemisphere midlatitudes at a similar latitude with different meteorological conditions but at different longitudes. Here, we address the question of whether similar correlations can be found there, although the meteorological situation might be different in terms of large-scale dynamics, i.e., in terms of developing baroclinic instabilities or evolving frontal systems. For this purpose, solely the reanalysis data are now used because of a lack of high-resolution radiosonde data with acceptable measurement quality for humidity variables.

Idar-Oberstein in central Europe (CE; 49.69° N, 7.33° E), which has already been highlighted in detail, has a low cyclone frequency with its maximum in the spring months (MAM). The second location is in central Asia (CA; 49.75° N, 87.25° E), with almost no cyclonic activity throughout the year. The third location is in the central USA (USA; 42.50° N, −86.5° E), where the cyclonic frequency for the region is high in winter (DJF) and spring (MAM) and low in summer (JJA) and autumn (SON). The fourth location is in the northern Pacific (NP; 49.75° N, −172.75° E), with very high cyclonic frequency through out the year except for a strong minimum in summer (Wernli and Schwierz, 2006).

As already investigated for the region of central Europe (Fig. 8), the TIL strength is also distributed among the different moisture classes for the other locations. The results are presented in Fig. 10 and confirm that the sTIL values tend to be higher when the average relative humidity with respect to ice is also high for the other regions. Also, the variability in sTIL increases with increased humidity in all regions. However, it is visible that the different regions exhibit different distribution shapes of the TIL strength. For example, the central USA shows the highest variability in the sTIL and the highest probability of strong sTIL events. In contrast, central Asia shows the lowest variability and the lowest probability of strong sTIL events.

To summarize, the co-occurrence of high values of humidity with strong values for sTIL is a robust feature of the Northern Hemisphere midlatitudes. The higher the averaged humidity is, the stronger the resulting TIL is.

## 3.4  Seasonal variations

This section deals with the seasonal variation of the tropopause inversion layer, which is also discussed in the literature. It was found that the interplay of water vapor on the static stability in the UTLS region occurs on seasonal timescales (Kunz et al., 2009; Hegglin et al., 2009).

### 3.4.1  Seasonal mean vertical profiles

Starting with the temperature profile for central Europe (Fig. 11a), the highest temperatures in the stratosphere and troposphere are found in the summer months; similarly, the lowest temperatures are found in the winter months. The autumn and spring months show similar temperatures in the troposphere and at the tropopause level but significantly differ in the stratosphere, with autumn temperatures being colder

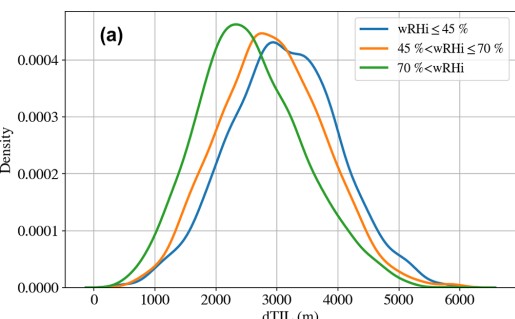
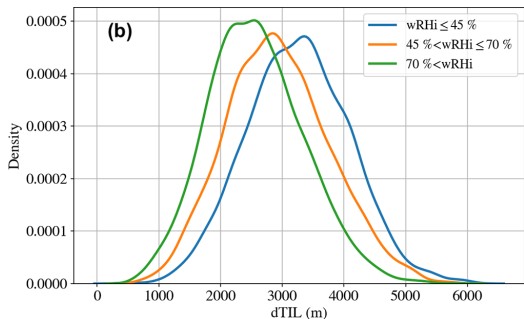

**Figure 9.** Probability density distribution for the radiosondes **(a)** and ERA5 **(b)** of the tropopause inversion layer depth (dTIL) for categories of average relative humidity over ice (wRHi). The low category (wRHi ≤ 45 %) is shown in blue, the medium category (45 % < wRHi ≤ 70 %) is shown in orange and the high category (70 % < wRHi) is shown in green.

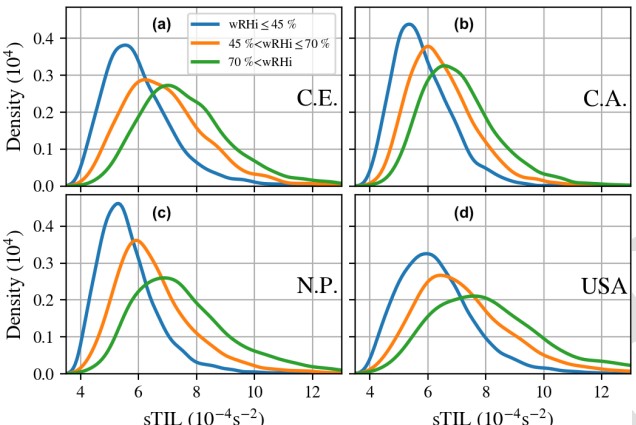

**Figure 10.** The probability density function of the TIL strength (sTIL) for average relative humidity categories. Central Europe (CE) **(a)**, central Asia (CA) **(b)**, the northern Pacific (NP) **(c)** and the central USA (USA) **(d)**.

than spring temperatures. For higher levels, the autumn temperatures match the winter temperatures.

The averaged RHi values for central Europe show only small seasonal differences in the troposphere. The most pronounced seasonal differences are found in the first kilometer above the tropopause. Based on the previous findings in Sect. 3.2.1, it is expected that the spring and summer months show the strongest TIL (high sTIL values) and the thinnest TIL (low dTIL values).

The static stability profiles show a peak of high static stability above the tropopause of similar magnitude $N^2 = 6 \times 10^{-4}\,\mathrm{s}^{-2}$ across every season, with a significant difference in static stability above. Within the first 3 km above the tropopause the static stability is on average $0.48 \times 10^{-4}\,\mathrm{s}^{-2}$ higher in spring compared to autumn and $0.22 \times 10^{-4}\,\mathrm{s}^{-2}$ higher in winter compared to summer, confirming previous findings by Schmidt et al. (2010) for the midlatitude region (40–60° N). The vertical profiles over central Asia (CA) (Fig. 11b) show weak seasonal differences for RHi and

the static stability. The summer and autumn months show slightly higher RHi below the tropopause. In contrast, the temperature profiles show a high variability throughout the year, with the highest temperatures in summer and the lowest temperature in winter.

The region of the northern Pacific shows small or even no differences in the temperature and RHi profiles between spring, autumn and winter (Fig. 11c). However, the summer season is clearly different, and the temperature profile is significantly colder in the tropopause region. At the same time, there is a strong increase in RHi above the first kilometer above the tropopause. This supports the idea that water vapor in the UTLS region has a cooling effect on the tropopause region (Randel et al., 2007; Randel and Wu, 2010). The static stability profile shows little seasonal differences, and only the autumn months have a smaller static stability peak.

The region of the central USA (Fig. 11d) shows the strongest seasonal differences in RHi and $N^2$. The RHi at the tropopause level takes on the highest values in spring followed by winter, summer and autumn in decreasing order. The peaks of $N^2$ show a similar pattern, with spring exhibiting the strongest peak, followed by winter, summer and autumn. The temperature profile is also different compared to the other regions, with summer exhibiting the lowest and winter the highest temperatures relative to the tropopause. By design the central USA location was introduced to include a region of frequent deep convection in the midlatitudes. There, significantly higher CAPE values are present in the summer months compared to the other regions (Taszarek et al., 2021), indicating greater deep convection activity. This leads to a stronger seasonal cycle in the vertical profiles in this region.

### 3.4.2 Seasonal cycle of TIL strength

The correlation between TIL strength and relative humidity is split seasonally using the PDF for different moisture classes for all four regions. The same moisture classes using the average relative humidity with respect to ice (wRHi)

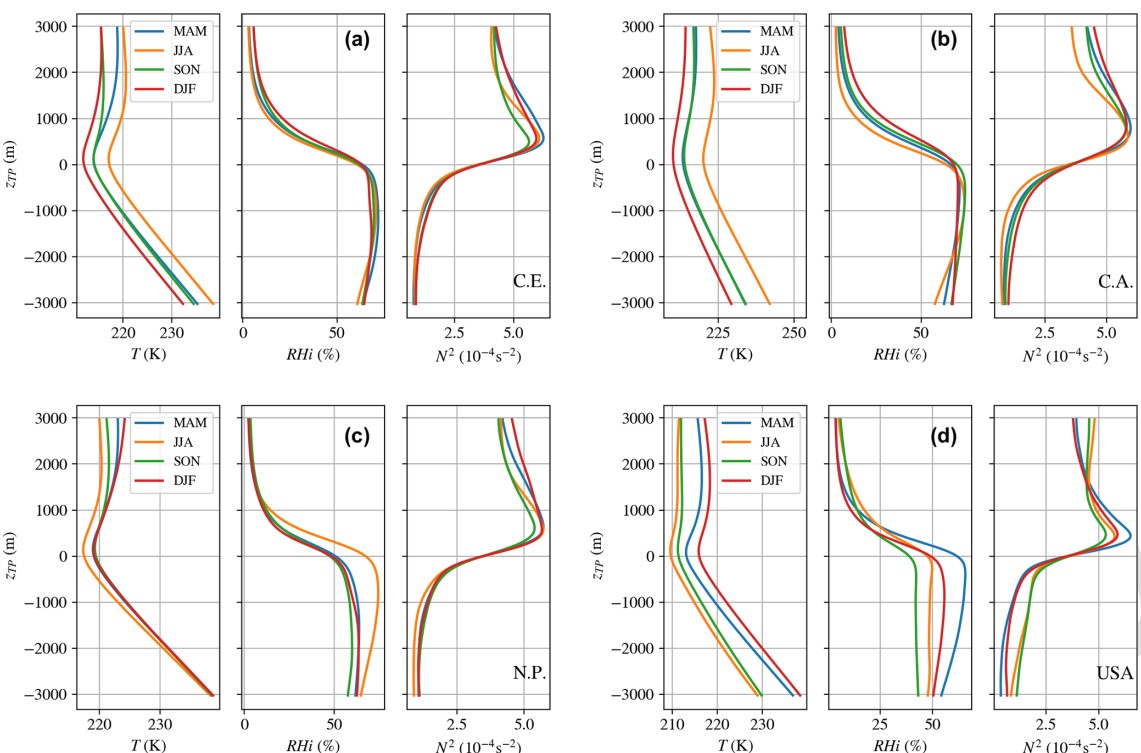

**Figure 11.** Mean vertical profiles of temperature ($T$), relative humidity over ice (RHi) and static stability ($N^2$) in the tropopause relative height ($z_\mathrm{TP}$). Colors indicate the different seasons, i.e., spring months (MAM) in blue, summer months (JJA) in orange, autumn months (SON) in green and winter months (DJF) in red. The panels represent the behavior for different regions; i.e., results for central Europe are shown in panel **(a)**, results for central Asia are shown in panel **(b)**, results for the northern Pacific are shown in panel **(c)** and results for the central USA are shown in panel **(d)**.

as in Fig. 8 are used. The correlation of higher TIL strength (sTIL) values with higher averaged values of wRHi is consistent across every season and every geographical location. Furthermore the correlation of higher sTIL variances with higher wRHi is also present in every region. Also, a distinct feature independent of the geographical region or season is that the low-wRHi category shows a mode between $5 \times 10^{-4}$ and $6 \times 10^{-4}\,\mathrm{s}^{-2}$, with a very low occurrence probability above $8 \times 10^{-4}\,\mathrm{s}^{-2}$.

The regions over central Europe (Fig. 12a) and the central USA (Fig. 12d) show a similar seasonal behavior. The highest values of sTIL (i.e., strongest TIL) are found in the spring months, while the minima are found in autumn. The winter months show higher variance and stronger extremes compared to summer, which show on average similar sTIL values to those in winter. The winter and summer similarity further supports the idea (according to Kunkel et al., 2016) that adiabatic processes (i.e., baroclinic forcings) and diabatic processes (e.g., radiative or latent heating) can have similar amplifying effects, although the timescales might not necessarily be the same. The evolution of baroclinic instabilities takes place within days, whereas radiative processes might act on timescales of up to few days (if the water vapor concentration is not changed drastically in between). How-

ever, cloud processes might act on much shorter scales (minutes to hours). Thus, a clear attribution remains difficult or even impossible. The main difference between central Europe and the central USA is the considerably higher variance of sTIL in the region of the central USA. Looking at the cyclonic frequencies, the region of the central USA shows a higher frequency than the region of central Europe across every season. A higher cyclone frequency alone does not explain the high variance in sTIL, as the northern Pacific does not have a higher variance in sTIL but has the highest cyclonic frequency of all the regions considered. This requires further research into the causes and the forcing mechanisms of the tropopause inversion layer.

The lowest variance values of sTIL are found in central Asia (Fig. 12b) across each season and humidity category. Wirth and Szabo (2007) suggested a strengthening of the TIL by baroclinic waves. The baroclinic waves are rare in central Asia as deduced from the low cyclonic frequency, which could explain weaker sTIL values compared to the other region, where baroclinic waves occur more frequently. Also, in central Asia the highest sTIL values are found in the summer months, which is more characteristic of the polar TIL as demonstrated by Randel et al. (2007) and Grise et al. (2010). This maximum in sTIL in the summer observation supports

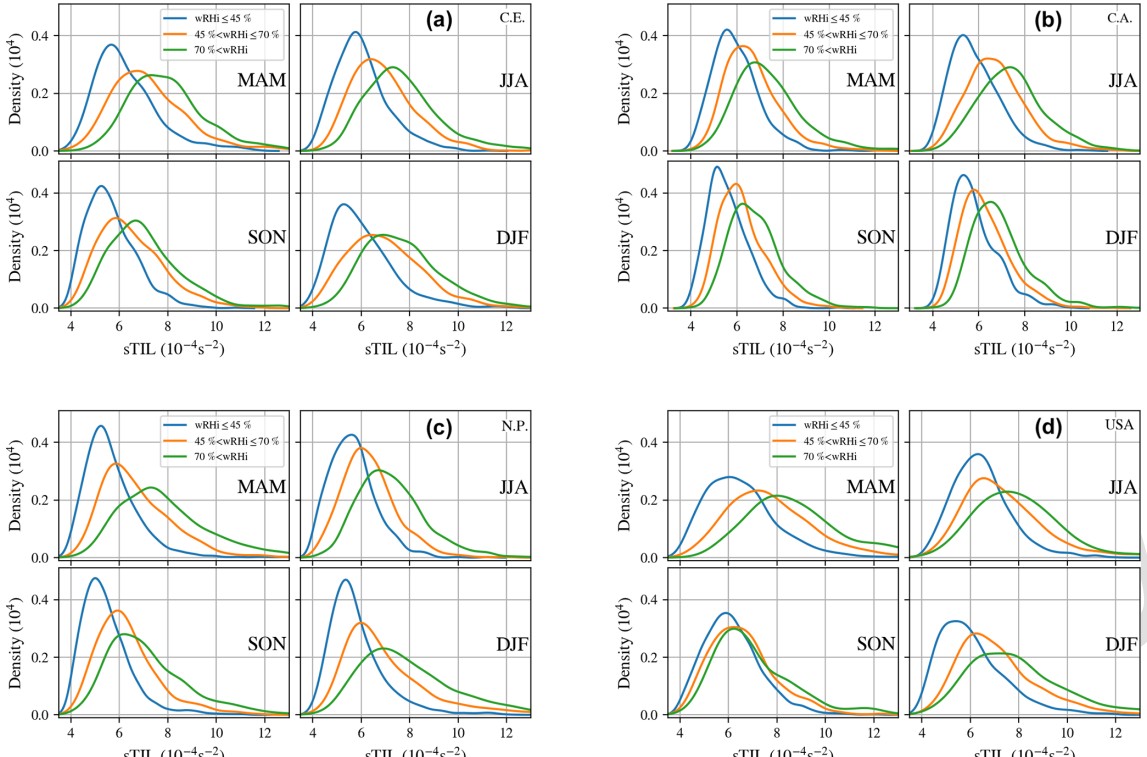

**Figure 12.** Seasonal probability density distributions of the tropopause inversion layer strength (sTIL) for categories of average relative humidity over ice (wRHi). Colors indicate the different wRHi categories: blue, low category (wRHi ≤ 45 %); orange, medium category (45 % < wRHi ≤ 70 %); green, high category (70 % < wRHi). The panels represent the behavior for different regions; i.e., results for central Europe are shown in panel **(a)**, results for central Asia are shown in panel **(b)**, results for the northern Pacific are shown in panel **(c)** and results for the central USA are shown in panel **(d)**. In addition, in each subfigure the season is indicated, i.e., spring months (MAM), summer months (JJA), autumn months (SON) and winter months (DJF).

the radiative mechanism suggested by Randel et al. (2007), since the forcing mechanism by baroclinic waves is reduced to a minimum for this region.

The region over the northern Pacific (Fig. 12c) shows a maximum of sTIL values in the winter and spring months and a similar minimum in autumn and summer. The maxima are most likely due the baroclinic wave activity, which has its maximum in spring and winter. It is noticeable that the mode in the wettest wRHi category is at higher sTIL values in spring than in winter. This suggests that water vapor and its radiative forcing amplify the sTIL in spring. The comparison between autumn and summer also shows an interesting aspect. The cyclonic activity has a significant minimum in summer and is lower in autumn than in spring and winter; nonetheless, spring and summer exhibit similar sTIL values. This gives the indication that the forcing mechanism through baroclinic waves and through the radiative effects might have a similar amplitude but may act on different timescales.

Altogether, consideration of seasonal and geographic differences reveals the overall robust relationship between the strength of TIL and average humidity with respect to ice in the upper troposphere (wRHi). However, if one looks more closely at the seasonal cycle in different regions with different synoptic characteristics, one finds different processes seen over the year. At the same time, evidence for different formation mechanisms for TIL (radiative forcing, baroclinic waves) can be found.

## 4 Conclusions

From theory and former model investigations, we can assume that the variable relative humidity over ice is a very meaningful quantity for investigations of properties of the tropopause inversion layer. Since the formation and sharpening of the TIL are first driven by adiabatic compression of the isentropes and are enhanced afterwards by irreversible diabatic processes, high values of relative humidity over ice are important markers for both occurring processes. For a detailed investigation, we used high-resolution radiosonde data over one German site (Idar-Oberstein; 49.69° N, 7.33° E) and the ERA5 data from the reanalysis project of the ECMWF (ERA5 grid point at 49.75° N, 7.25° E). In a first step, we could show that both data sets agree very well and are able to represent the TIL and its features quantitatively and qualita-

tively. In a second step, the properties of the TIL in comparison with some mean relative humidity measures are investigated. As a major result, we find that sharper or stronger TILs occur on the one hand for higher RHi values, which points to the adiabatic contribution (i.e., adiabatic lifting of the moist layers), and on the other hand also for sharper gradients of RHi at the top of the moist layer, indicating diabatic processes (e.g., radiative effects). These results are very robust through seasonal variations, as obtained from further analysis of the ERA5 data. In addition, the connection between high humidity values and strong TILs can be found at other midlatitude geographical regions, namely the central USA, central Asia and the northern Pacific, at approximately the same latitude throughout all seasons. Again, variations in the strength and moisture values might be related to the strong activity in large-scale flows, i.e., in baroclinic instabilities and frontal systems.

Overall, the ERA5 data are well suited for this kind of investigation, and the qualitative features of TILs are well represented. The connection between relative humidity and TIL features is robust and corroborates the former findings in model studies by Kunkel et al. (2016), who stated that diabatic and adiabatic processes can have similar amplifying effects on the TIL. Although it is not possible to discriminate the adiabatic and diabatic components in the analysis, we could show that the relative humidity is well suited for this kind of analysis, pointing to the relevant processes for the evolution of the tropopause inversion layer. Possible future investigations of the TIL using the ERA5 data set might concentrate on the Lagrangian evolution of TIL structures.

**Code availability.** Code for the data processing and analysis is provided on Zenodo (https://doi.org/10.5281/zenodo.10604349, Köhler, 2024).

**Data availability.** The radiosonde data are publicly available from https://opendata.dwd.de/climate_environment/CDC/observations_germany/radiosondes/high_resolution/historical/ (DWD, 2024). TS6 The ERA5 data are available from the Copernicus Climate Change (C3S) climate data store (CDS) at https://doi.org/10.24381/cds.adbb2d47 (Hersbach et al., 2023).

**Author contributions.** DK, PR and PS designed the study; DK carried out the data analyses; DK, PR and PS contributed to interpreting the results and writing the paper.

**Competing interests.** The contact author has declared that none of the authors has any competing interests.

**Disclaimer.** Publisher's note: Copernicus Publications remains neutral with regard to jurisdictional claims made in the text, published maps, institutional affiliations, or any other geographical representation in this paper. While Copernicus Publications makes every effort to include appropriate place names, the final responsibility lies with the authors.

**Acknowledgements.** We thank Deutscher Wetterdienst (DWD) for providing the high-resolution radiosonde data and the European Centre for Medium-Range Weather Forecasts (ECMWF) for providing the ERA5 reanalysis data. We thank Daniel Kunkel for fruitful discussions and for getting the data from ECMWF. We also thank Peter Hoor for fruitful discussions. Philipp Reutter acknowledges support by the Deutsche Forschungsgemeinschaft (DFG) within the Transregional Collaborative Research Centre TRR 301 TPChange, project ID 428312742, project C1. Peter Spichtinger acknowledges support by the DFG within the Transregional Collaborative Research Centre TRR 301 TPChange, project ID 428312742, project B7. The study contributes to the project Big Data in Atmospheric Physics (BINARY), funded by the Carl Zeiss Foundation (grant P2018-02-003). The authors thank the two anonymous reviewers for their constructive comments and suggestions that improved the paper.

**Financial support.** This TS7 research has been supported by the Deutsche Forschungsgemeinschaft (DFG) collaborative research program "The Tropopause Region in a Changing Atmosphere" TRR 301 (project ID 428312742).

This open-access publication was funded by Johannes Gutenberg University Mainz.

**Review statement.** This paper was edited by Martina Krämer and reviewed by two anonymous referees.

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

## Remarks from the language copy-editor

CE1     Please confirm.

CE2     Please confirm change (deleted "pressure" as requested) or clarify.

CE3     Please verify and confirm the changes to the quote below.

## Remarks from the typesetter

TS1     Please confirm.

TS2     Changes in values would require editor approval (0.5 to 1). Please provide an explanation regarding this correction that can be forwarded by us to the editor. Thank you very much in advance.

TS3     Please confirm.

TS4     Please confirm citation.

TS5     Please confirm.

TS6     Please confirm changes made to this section.

TS7     Please confirm this section. If the sentence "The subprojects of the research program are provided in the acknowledgments." should have also been added at the end, please let me know.

TS8     Please confirm.

TS9     Please confirm.