# Peer review of "Relative humidity over ice as a key variable for Northern hemisphere extratropical tropopause inversion layers"

_EGUsphere, 2023_

## Referee Comment (RC2)

*Review of*

**"The extratropical tropopause inversion layer and its correlation with relative humidity "**

*by Daniel Köhler et al.*

**General**

This is a good paper. It addresses the formation mechanism of the extratropical tropopause inversion layer (TIL) and the different forcing mechanisms of the TIL discussed in the literature (Randel et al., 2007; Wirth and Szabo, 2007). Here baroclinic waves and radiative ($H_2O$) processes are relevant, but the time scales involved should not be ignored (and they might be different for the two mechanisms). Does the analysis provided here allow statements about radiative/dynamical time scales?

The paper is based on high-resolution radiosonde data from Idar-Oberstein (but give location (lat/lon) at first mention, and the relevant time period) with ERA5 data. This is good. However, I would emphasise differences as well a similarities between ERA5 and the sondes (see below).

The paper then moves on to investigate the influence of relative humidity with respect to ice (i.e. $H_2O$) on the extratropical TIL. Further, based on ERA5, longitudinal and seasonal variability of the TIL is discussed. But the paper should be quantitative and more accurate (and less vague) than saying something like ". . . reveal consistent relationships in various extratropical regions of the Northern Hemisphere under different meteorological conditions".

I am not sure if the authors agree with the assessment in this review – they do not need to do so. But a much clearer message of the paper would be very helpful. This is not clear from the present draft – in particular not in the title and the abstract. I think the paper would be more valuable if the message would be much clearer in a revised version.

Finally, while I am mentioning papers here that might be potentially of interest (and some already cited), I am certainly not suggesting the citation of particular papers.

Overall, I think this is a good helpful paper of interest to the readership of ACP. I

suggest a major restructuring to make the key points of the paper clearer and more accurate in a revised version.

**Comments in detail**

**Abstract and title**

There are guidelines for ACP papers, in particular the title, abstract, and concluding section:

`https://www.atmospheric-chemistry-and-physics.net/policies/guidelines_for_authors.html`

Titles should be concise and consistent with the content and purpose of the article. For research articles, ACP prefers titles that highlight the scientific results/findings or implications of the study. Abstracts should have fewer than 250 words – I think the paper can be improved in this respect.

The paper should be very clear what the main findings are and what the advance of knowledge of the study is.

**Comparison of ERA5 and a radiosonde station**

As I understand the paper, the basis of the paper is a comparison between the TIL in ERA5 and in the data from a radiosonde station. After a 'validation' of the ERA5 data with the radiosonde data, further conclusions for the TIL in the latitude range of the station are drawn.

This is good, but the paper is not very clear about this. The stations is called "Idar-Oberstein", sometimes only "Germany" is mentioned, the period of radiosonde data is often not mentioned, on other occasions the latitude/longitude of the station is mentioned – all the information is in the paper, but the reader should not be forced to search the entire paper to find the necessary information.

Most importantly, as I read the paper the results are relevant for northern hemisphere mid-latitudes (close to $50°N$) – is this correct?. If the authors agree, this fact should be evident in the paper, in particular abstract and title.

**Tropopause**

The entire concept of the TIL is based on using tropopause relative coordinates. Yes, this is reported on page 7 of the manuscript, but I suggest making this concept clear upfront. Further, determination of the tropopause is not straightforward (e.g., in ERA5), there is a an extensive discussion in a recent publication (Hoffmann and Spang, 2022); this publication also addresses the issue of a fixed pressure grid and different interpolations, which might be helpful here.

Moreover, on page 6 of the manuscript, the classic WMO definition of the lapse rate tropopause is cited, however this definition leaves room for interpretation (Maddox and Mullendore, 2018). Exactly which definition of the lapse rate tropopause has been used? As stated in the paper the classic WMO definition is old and does not necessarily take into account the use of more recent gridded and high resolution data (see e.g., Reichler et al., 2003).

The authors mention the review by Gettelman et al. (2011), which is good. However, there are also other reviews of the determination of the tropopause (Hoinka, 1997) and there is also a tropopause definition based on isentropic potential vorticity gradients (Kunz et al., 2011).

**Figs. 1, 2 and 3**

First, I find these figures very helpful, they explain the concepts used here before more general statements are made.

However, I suggest that the scheme in Fig. 3 is closer to reality (Fig. 2); the static stability is not constant wit altitude (above 11.25 km) and the kink at 11.25 km in Fig. 3 is not seen in the real data (Fig. 2).

Further, I like Fig. 2 a lot, but I cannot see why an interpolation to a fixed altitude grid is necessary to produce the figure – doesn't the interpolation introduce an unnecessary smoothing? Most importantly, likely, the difference between sonde and ERA5 that I see in Fig. 2, is an important result. If I were an author, I would flag this result more strongly and more quantitatively in the paper (e.g. abstract, conclusions).

**Equation 8**

In Equation 8, a measure is defined for the deviation between ERA5 and the sondes. However, this definition is not unique. It is a choice, isn't it? The problem I see is that deviations between $E$ and $R$ could cancel out when integrated over a certain altitude range. That is locally there could be a substantial deviation between $E$ and $R$, but $\bar{D}$ could be rather small, depending on how the range $z' - z_0$ was chosen. Why are no absolute values considered of the deviation between $E$ and $R$?

**Minor issues**

- l. 10: I would not use the term "strong agreement" when two temperature profiles (say) are very similar.

- l. 10: "geographical": what is meant here is the longitudinal variation. Correct?

- l. 18: "distinct and intriguing feature known as the tropopause inversion layer": here it would be helpful to report particular features of the TIL, rather than saying "intriguing". What are the most relevant properties of the TIL? Such information comes later in the introduction, but it would be good to have this up front.

- l. 25: why "inert substances" ? Isn't the TIL a barrier for vertical transport even if substances are not chemically inert?

- "hypothesis " should be plural

- l. 51: give latitude and longitude of Idar-Oberstein

- l. 52: radiosonde data (not "sondes")

- l. 69: focuses

- l. 72: these thresholds look somewhat arbitrary. Are there citations? Are there any indications in the household data? Would a temperature of (say) 450 K be okay?

- l. 74: units should not be in italics

- l. 80 latest $\longrightarrow$ most recent

- l. 83: state explicitly how many levels were used. State the top altitude used. Also the approximate vertical resolution in ERA5 here would be useful to report.

- l. 85: "closest grid point" – this is always the same point in the ERA5 grid – correct? This point could explicitly be mentioned.

- l. 90: here and elsewhere "the data sets" is used, but is must be made clear that ERA5 and radiosonde is meant. It is likely better to err on the safe side and explicitly state what is meant.

- l. 91: the "improved statistical analysis" is not obvious from the paper.

- l. 92: "height" be more precise here, geometric altitude, pressure altitude, geopotential altitude etc., is not the same thing and not available in each data set.

- l. 94: citation for the buoyancy speed?

- l. 96: what is the argument for cubic spline?

- l. 99: "inconsistent with respect to time" is unclear.

- l. 99: Which "data set"?

- l 110: citation for this statement?

- l. 117: quantify "slightly"

- Eq. (2): I am not familiar wit this approximation; is there a citation? Or an explanation of Eq. (2)? How accurate is Eq. (2)?

- Eq. (4) is stated here that as an approximation for dry air: this aspect should be made clear here. Make clear what the issue is of wet vs. non wet conditions.

- Eq. (5): I suggest to make clear (here and elsewhere) what $z$ is – is it geopotential height in Eq. (5)?

- l. 155: see also other work (e.g., Reichler et al., 2003; Maddox and Mullendore, 2018) on using the classic WMO tropopause definition for modern, gridded data.

- l. 157: be clear about which data sets, "both" is a bit vague

- l. 169: which period of the radiosonde data?

- Fig. 2: This figure is good. I suggest adding some discussion in how-far the sondes and ERA5 do *not* agree.

- l. 172: "Another"? this paper is on the TIL.

- l. 175: Suggest stating the importance of the TIL earlier in the paper (introduction).

- l. 177: How is the value of 3 km chosen?

- l. 182: say which features.

- Eq. 7: $p500_z$ is an altitude (not a pressure) – the symbol is confusing.

- sec. 3.1.1: be clear what is compared with what.

- l. 211: stating (e.g.) "upper troposphere" is not enough here, the exact range $z' - z_0$ that was used should be reported.

- l. 220: are $\longrightarrow$ is

- l. 223: quantify "thin"

- l. 226: do you really want to give three significant figures here?

- l. 241: is this statement consistent what is shown in Fig. 2?

- l. 244: "averaged" over which region?

- l. 269: I agree, but Fig. 2 also shows the limitations.

- l. 272: which "data"?

- l. 275: Unclear sentence, two times wRHi?

- l. 275: give the Fig. where the PDF can be seen

- l. 286: report the latitude range for which the comparison is valid.

- l. 289: resolution or vertical resolution?

- l. 291: again, point to the figure in question here, merge wit the next sentence (we show)...

- l. 293: "the TIL depth dTIL is shifted to lower values" – this is clear from Fig.9. But then (l. 294) "the depth of the TIL is always in the range" – so does DTIL change with humidity or not? Where would I see the "second mode"? In 9b? More help for the reader? I find this discussion somewhat confusing here.

- l. 296: it is not clear why there is an "artefact" here and what this implies.

- Fig. 9; mention a and b in the caption.

- l. 302: "sharpening the TIL ... depth" – do you have an argument why?

- l. 306: state here immediately which three regions. I think the main point is longitudinal variation here.

- l. 328: 328: "to act" $\longrightarrow$ "acts"

- l. 337: give section/figures for "previous findings"

- l. 340: quantify the "differences" found here

- l. 348: is this true? I do not see the strong increase in RHi in Fig. 2

- l. 354: "could be due" sounds rather speculative.

- l. 366: do both "effects" have similar time scales? Would this not be important?

- l. 373: compared

- l. 375: confused about "polar" and "summer" in this sentence

- l. 378: maximum in what?

- l. 381: what are the time scales in question here? Should not be forgotten.

- l. 384: amplitude yes, but what about time scales?

- l. 386: really "upper troposphere"? That means below the tropopause?

- l. 387: "different" sounds vague here.

- l. 391: give latitude and longitude of Idar-Oberstein, give length of the time period

- l. 393: same location is not clear; ERA5 does not have a grid point at Idar-Oberstein.

- l. 398: this is not important, but meteorologists typically talk about high temperatures, not warm temperatures.

- l. 401: at which altitude?

- l. 401: "too stable": compared to what?

- l. 418: provide the code on a web-page, not only "upon request ".

- l. 420: It would be good to report not only the raw data, but to also create a location where the processed data of this study are available (say TIL strength).

- l. 427: give correct "spelling" of ECMWF

- l. 479: correct authors list?

- l. 490: give page rage for the citation.

**References**

Gettelman, A., Hoor, P., Pan, L. L., Randel, W. J., Hegglin, M. I., and Birner, T.: The extratropical upper troposphere and lower stratosphere, Rev. Geophys., 49, RG3003, https://doi.org/10.1029/2011RG000355, 2011.

Hoffmann, L. and Spang, R.: An assessment of tropopause characteristics of the ERA5 and ERA-Interim meteorological reanalyses, Atmos. Chem. Phys., 22, 4019–4046, https://doi.org/10.5194/acp-22-4019-2022, 2022.

Hoinka, K. P.: The tropopause: discovery, definition and demarcation, Meteorol. Z., 6, 281–303, 1997.

Kunz, A., Konopka, P., Müller, R., and Pan, L. L.: Dynamical tropopause based on isentropic potential vorticity gradients, J. Geophys. Res., 116, D01110, https://doi.org/10.1029/2010JD014343, 2011.

Maddox, E. M. and Mullendore, G. L.: Determination of Best Tropopause Definition for Convective Transport Studies, J. Atmos. Sci., pp. 3433–3446, URL `https://doi.org/10.1175/JAS-D-18-0032.1`, 2018.

Randel, W. J., Wu, F., and Forster, P.: The extratropical tropopause inversion layer: Global observations with GPS data, and a radiative forcing mechanism, J. Atmos. Sci., 64, 4489–4496, https://doi.org/10.1175/2007JAS2412.1, 2007.

Reichler, T., Dameris, M., and Sausen, R.: Determining the tropopause height from gridded data, Geophys. Res. Lett., 30, 2042, https://doi.org/10.1029/2003GL018240, 2003.

Wirth, V. and Szabo, T.: Sharpness of the extratropical tropopause in baroclinic life cycle experiments, Geophys. Res. Lett., 34, L02809, https://doi.org/10.1029/2006GL028369, 2007.

---

## Author Comment (AC1)

**Reply to Reviewer**

We thank both referees for their insightful reviews and helpful and constructive comments. This improved the quality of the manuscript.

We have taken the suggestions and criticism as an opportunity to structure the manuscript more clearly. In doing so, we have concentrated on explaining in more detail why we examine the TIL together with the relative humidity with respect to ice. This is expressed, among other things, in a changed title as well as in a detailed discussion of relative humidity w.r.t. ice as an important parameter when analyzing the tropopause region. We also included a discussion on different tropopause definitions.

In the following, we answer to the comments point by point. *Questions and remarks of the reviewers are marked in orange,* reply of the authors are marked in black and *changes to the manuscript are marked in blue.*

**Reviewer #1**

The paper deals with the correlation between TIL properties and relative humidity in a few kilometers thick layer around the tropopause. The authors mention two theories or hypotheses that explain the strong stability maximum that is found in high-resolution profiles at the tropopause, namely a dynamic and a radiative mechanism. Their results seem to support both theories, but not always simultaneously at the same location and season.

It is not clear to me whether supporting one or both of these theories was the motivation of the study; this is not clearly stated. And it seems to have been not so clear to the authors as well, as it is never clear what the correlations actually mean. Sometimes the text seems to indicate that high relative humidity CAUSES a strong (but thin) TIL, but a corresponding mechanism is not mentioned. Furthemore, if the radiation mechanism is in focus, it should be the radiation extinction and emission by the water molecules that is important, that is, it is the absolute rather than the relative humidity one should look at. If, however, the dynamical mechanism is in focus, consideration of the relative humidty is probably the correct choice since vertical motions lead to increasing relative humidity.

In any case, I miss the description of mechanisms, that explain or at least suggest how water molecules, either in absolute number or as relative humidity, are involved in the formation of the TIL by either radiation or dynamics.

The paper thus needs a major revision before it can be accepted for publication. Below I give some further suggestions for improvements.

Major points:
Section 3.1.1:
Why do you use the simple differences E(x)-R(x)? This may average out to quite small values and you think everything is ok, where in reality you might have large local differences. Usually, therefore, other measures are used that avoid such cancellation effects, e.g. RMSE, mean ABSOLUTE difference etc. I think, to convince the reader, it is necessary that these traditional standard measures are used instead of the simple average distance. This will certainly broaden your histograms in figures 4 and 5, and perhaps modify or change your conclusions.

We have replaced the difference plots with plots of the average absolute difference based on the new equation 8.

$$\overline{D}_{\text{abs}}(\chi) := \frac{1}{z' - z_0} \int\limits_{z_0}^{z'} |E(\chi) - R(\chi)| \mathrm{d}z \approx \frac{1}{z' - z_0} \sum_{z_0}^{z'} |E(\chi(z)) - R(\chi(z))| \Delta z \qquad (8)$$

This affects the distributions of the deviations, i.e. Figures 4,5 and 6, and the interpretation of the results. The text was partly rewritten. We also added following discussion to the manuscript:

*We chose a metric including absolute values of differences in order to avoid undesired cancellation effects of positive and negative contributions. In this sense, we used a metric inspired by the L1-norm. Thus, the resulting distributions are expected to be skew and might have (exponential) decaying tails. The resulting data set is visualized in a probability bar chart and the corresponding median, mean and standard deviation for the different variables are presented in figures 4, 5, and 6, respectively.*

Lines 218 ff: another difficulty arises from averaging over the profiles: In this paragraph a shift to higher values of RHi in ERA5 is explained with details of the humidity gradient at the TP. While this is a plausible explanation, it cannot be seen from averaged profiles. Here it would help to split the data into data below the TIL and above the TIL. Finally, please think whether the quoted values are actually as precise as given (e.g. 10.318).

We revised this investigation of the data; the newly introduced metric with absolute values of differences does not allow some interpretations as stated in the former version. However, the data is already split into tropospheric and stratospheric data, leading to different deviations. It would be difficult or even impossible to split the data in "above" and "below" the TIL, since it is not clear, which reference level should be taken into account.

We reduced the precision of the given values to two decimal places.

Section 3.2.1: I see that the mean T-difference at the TP between strong and weak TIL cases is about 10K, and this difference is almost constant throughout the 3 km below the TP. If a strong TIL has a higher TP, one can estimate a height difference of 1-1.5 km (according to usual lapse rates). This means, the T-difference applies in the whole profile at least 1.5 km down from the TP. Your argument in line 257, that there should be a stronger T-gradient, is weak, because it seems that the T-gradient (on average) is the same in all TIL-classes. A similar observation can be made for the RHi profiles. The higher RHi in the strong TIL case is already present at least 1.5 km below the TP. So the question here is, why and how the TIL is affected by profile characteristics that begin at least 1.5 km below the TP. You should also consider the absolute humidity, q. From what I see, I expect that a strong TIL is correlated to the lowest values of q and vice versa. This might be important for the question how radiation can affect the TIL. The argument, that high RHi at the TP may result from vertical motion, is reasonable, but it weakens your statement from above that radiation might cause the sharpness of the TIL. I suggest to go back and to downplay this argument. This would also strenghten the justification to focus on relative than on specific humidity (which does not change in vertical motions).

We agree that the interpretation of the colder temperature profile in terms of a stronger or weaker TILs is not very strong. We have added some text to clarify this. However, this is not the major result of the study; the correlation between RHi and TIL strength is the more relevant result, which is not affected by this interpretation.

We added a detailed discussion in the introduction, why we think that the relative humidity with respect ice is well suited for this investigation:

*In a recent model study, Kunkel et al. (2016) were able to show that the formation of the TIL is probably driven by a combination of different adiabatic and diabatic processes. In a first step, evolving baroclinic instabilities lead to a compression of isentropes, which in turn results into sharper gradients of the stability. Relevant processes in this respect are horizontal convergence in anticyclonic regions, strong upward motions, e.g. triggered by convective instabilities, and gravity waves triggered by the large scale flow, respectively. Since these*

*changes are mostly adiabatic and thus reversible in general, in a second step diabatic processes as turbulence/mixing, cloud formation and resulting latent heat release and radiative heating/cooling by trace gases (as water vapor) modify the TIL irreversible. As showed by Kunkel et al. (2016), the diagnostics of these processes is quite difficult and from measurements it might be quite impossible to disentangle the contributions of the different processes.*

*However, by careful inspection of the scenario, there is a quantity, which might be considered as a proxy for these different processes.*
*The relative humidity (with respect to a stable phase of water, i.e., liquid or solid) is the control variable for many cloud processes. On the other hand, this variable, as combined by the mass concentration of water vapor, pressure, and temperature, is a good indicator for adiabatic expansion processes (i.e. cooling); high values of RH can be expected if moist air is adiabatically lifted. In the tropopause region within the low temperature regime (i.e. T< 235 K), relative humidity over ice (RHi) is the relevant quantity, since solid ice is the stable phase there. Thus, RHi might be a good indicator for the strong lifting of air masses in baroclinic instabilities, thus working as a proxy for strong TILs.*

*Water vapor is a strong greenhouse gas, especially in the infrared range; the absolute concentration of water molecules controls the amount of emission and absorption. Particularly in case of a moist layer we would expect a strong emission of energy in the infrared spectrum, and thus a cooling of the layer. However, the total amount of water molecules is not the only reason for strong emissions. Since the atmosphere is layered, the concentration of water vapor in adjacent layers is also of importance. If the layers of different temperatures have a similar amount of water vapor, the emitted radiation is easily absorbed by the layers on top. Thus, a strong gradient in concentration (e.g. a layer with low concentration on top of a layer with high concentration) leads to a much stronger cooling rate than in a situation with weak gradients. Since it is difficult to measure (or determine) the gradient in water vapor concentrations, a good compromise is the use of relative humidity. Since it is linear in the water vapor concentration, it represents the gradients in a meaningful way. Because of small temperature changes in adjacent layers with strong vapor gradients, the impact of the temperature is quite negligible. For the tropopause region, this phenomenon was investigated in a study by Fusina and Spichtinger (2010); a stronger gradient in RHi leads to a much more pronounced cooling on top of the moist layer.*

*In summary, the use of relative humidity over ice in the tropopause region might help to detect strong TILs. Or in other words, correlations between high values of RHi and strong TILs would corroborate the two step formation of TILs with adiabatic and diabatic components. Therefore the use of RHi is highly relevant for the investigation of the tropopause inversion layer.*

**Minor points:**
Line 67: Why explains the latitude of Idar Oberstein that the geopotential height is close to the geographic height?
Due to the greater acceleration by gravity at the poles, the geopotential at the same altitude is greater there than at lower latitudes. At a latitude of 45°N, the geopotential height corresponds to the geometric height. We therefore find it sufficient to apply this assumption to a latitude of 50°.

L 70: Why are profiles discarded when they end before 20 km? The tropopause in Idar Oberstein probably never reaches such an altitude.

As we are not only looking at profiles at the Idar-Oberstein site in this study, but also in other regions, we wanted to use a uniform threshold value in order to include individual cases with higher tropopause levels in the statistics. Since the determination of the parameters requires that data must still be available at least 3 km above the thermal tropopause, we think that 20 km is justified as a minimum requirement for a homogeneous data set. Usually the radiosondes can reach this altitude range without problems, so it is not a major limitation.

L 77-78: Are the Miloshevich corrections applicable for the RS92 and RS41? And how large are the corrections on average?

The corrections are applicable for the RS92. Due to the small subset of RS41 ascents, we have treated the profiles as equivalent to the RS92 ascents. Since the differences in the extratropics between the two probe types are very small (https://www.vaisala.com/sites/default/files/documents/RS-Comparison-White-Paper-B211317EN.pdf), we think the application of the correction is justified. The corrections are in the order of 5% in humid and 20% in dry conditions.

L 80ff: This sentence is a bit misleading. It is not "past model forecast" but rather recalculation (hindcast) of the past weather with a recent version of the forecast model. Further, the vertical dimension is represented by rather than calculated on sigma coordinates.

You are right. We reformulated the sentences in the manuscript

> *ERA5 is the most recent reanalysis product of the ECMWF (Hersbach et al., 2020). The reanalysis is a mix of a recalculation of past weather with one fixed forecast model version (IFS CY41R2) and assimilated measurements made for each available time. The high resolution data set has a horizontal resolution 0.25° in longitude and latitude. The vertical dimension of the atmosphere is represented by hybrid sigma/pressure (model) levels in ERA5 (ECMWF, 2020), the number of levels is 137 of which only levels up to the lower stratosphere are used.*

L 106 ff: see above. If it is cloud formation, then RHi is the natural variable to choose. However, if it is radiation, it should be the absolute water vapour concentration that regulates the optics.

We would like to refer to the extended discussion on the use of RHi in the introduction. See also reply to your comment on Section 3.2.1

Section 2.2.1: I am not sure whether this section is actually needed. The equations are textbook knowledge, and it probably suffices to write a few sentences instead of a subsection. The approach for the numerical treatment of the derivation should, however, be retained, as this is essential for the paper. The other details are not actually needed here (they might be important elsewhere, though). The lead away from the topic of the paper.

We think that this section makes sense because of the reproducibility. As shown in Murphy and Koop (2005) or Baumgartner et al. (2020), it is important which parameterizations are used for the calculation of essential quantities such as saturation vapor pressure or heat capacity, even if the deviations in the specified temperature and humidity range appear small.

L 153 ff: This remark should be removed. First, the reader cannot see what in the WMO definition requires to look "broadly" or at large scale at the system, and second, the potential effects of this "tacit" assumption are then ignored, for whatever reason. This puzzles the reader and leads away from the central topic.

We added a discussion on tropopause definitions in the new section 2.3.1. For the sake of readability, we will not show this section in this document. The reader is referred to the manuscript.

L 165 ff: For WHICH averaging process? It comes out of the nothing. I have no clue what happens here.
AND
L 168 ff: and it does not get clearer. It seems that you average your 10000 or so profiles, but that is not so clear to me. Here is the place to provide more details.

We tried to clarify, why we do the averaging and rewrote this part in the manuscript

> *In order to be able to compare the large number of radisonde data and ERA5 profiles, a tropopause-centered coordinate system was introduced. For this purpose, the thermal tropopause is identified in each radiosonde and ERA5 profile for Idar-Oberstein and defined as the tropopause height $TP_z$. This means that all profiles now live in the same coordinate system and can be averaged to obtain mean profiles of temperature, humidity and static stability. Therefore. we introduce a new height variable $z_{TP} = z - TP_z$ (6)*
>
> *relative to the tropopause height $TP_z$ (as derived by the WMO criterion, see above). Negative altitude values $z_{TP}$ denote the upper troposphere, whereas positive altitude values $z_{TP}$ represent the lower stratosphere. For the averaging process the single profiles are transformed into the $z_{TP}$ coordinate system and the arithmetic mean of a meteorological variable $\chi \in \{temperature, relative\ humidity, static\ stability\}$ is calculated, summing over all profiles at a certain height.*
>
> *The mean profiles relative to tropopause height of temperature, static stability ($N^2$), and RHi can be seen in Figure 2 for the radiosonde measurements (black) and the corresponding ERA5 data set (red) at the location of Idar-Oberstein. Even for the mean profiles the characteristics of the TIL, i.e., the strong increase in $N^2$ at around $TP_z$ can be seen clearly, as described in the next section.*

L 236 ff: Please reformulate. I think, one should not write that the LS is "more unstable" ... or "too unstable". The word "unstable" should be avoided when it is about the LS.

We agree. We used "less stable" instead of "more unstable" throughout the manuscript.

L 301 ff: If you write "a moist upper troposphere is sharpening the TIL", I think you must at least have a mechanism for the very process in mind and you must tell the reader how it works. But if you only find a coincidence (that is, by the way, a better word describing what you have than correlation, which has a mathematical definition), you should only write that there is a coincidence and avoid the impression that there is a certain process causing high relative humidity to sharpen the TIL.

We agree and rewrote this part.

> *In summary, it can be seen that the drier the air is in this region, the thicker the TIL is. In other words, a moist upper troposphere coincides with a decreased depth of the TIL. In combination with the findings from the section before, we can conclude, that for a more humid upper troposphere, we can find a stronger but vertically more confined TIL feature.*

L 324/325: Again, please reformulate. You found a coincidence, but nothing shows that the humidity has an influence on the TIL.

We agree and rewrote this part.

> *To summarize, the co-occurrence of high values of humidity with strong values for sTIL is a robust feature of the extratropics.*

L 327-329: Check the sentence. It sounds ugly. And it again indicates the RH somehow acts physically on the TIL.

We have reformulate the sentence.

*This section deals with the seasonal variation of the tropopause inversion layer, which is also discussed in the literature. It was found that the interplay of water vapor on the static stability in the UTLS region occurs on seasonal time scales (Kunz et al., 2009; Hegglin et al., 2009)*

L 347-348: If water vapour has this cooling effect, shouldn't then the absolute humidity be high? Can you show this?

Generally, it is the total amount of water molecules which determines the absorption and emission of infrared radiation. If the temperature variation is small, then relative humidity is directly proportional to the water vapor concentration, thus RHi is directly linked to the emission/absorption. However, if adjacent layers are investigated, the vertical gradient plays a major role (see Fusina & Spichtinger, 2010). Even in this case, RHi can be used for the quantification, since again the temperature variation is small.

L 365/366: "The winter/summer similarity further support the idea that radiative and baroclinic forcing can have similar amplifying effects." Please explain this statement.

Due to the lower baroclinic activity, the radiative effect is dominant in summer, while the opposite is the case in winter. Despite this difference, there is a similar enhancement of TIL with higher humidities, which is more likely to be attributed to the radiative mechanism in summer, while baroclinic activity is more likely the cause in winter.

L 413: the word "confirm" is too strong. I would say your data do not contradict these hypotheses and would be expected in one or both of the scenarios.

We agree and reformulated the sentence.

*The regional and seasonal analysis also showed the indication of the two suggested forcing mechanisms, the dynamical forcing by Wirth and Szabo (2007) and the radiative forcing by Randel et al. (2007).*

L 415 and L 417: "this is supporting...", again I think, this is too strong and should be reformulated.

We respectfully disagree, we think that the wording reflects our interpretation quite well.

Miscellaneous:
Line 26: hypotheses
We corrected this in the manuscript.

L 27: analyses, a stronger
We corrected this in the manuscript.

L 59: radiosonde soundings
We corrected this in the manuscript.

L 69: focusses
We corrected this in the manuscript.

L 72: replace "unscientific" with "unrealistic". The threshold values still appear quit high. Would 400 K be more realistic, or 200% RH?
We rewrote this part to

> *419 profiles are discarded, 311 due to insufficient maximum height, 19 due to missing data*
> *and 89 due to unreliable values of temperature and relative humidity.*

Actually, we leave the exact thresholds out, since the nonphysical values are such high that they just exceed all relevant values. Thus, it is not really important which exact value is used.

L 86: radiosonde (twice)
We corrected this in the manuscript.

L 88: What is the difference between PHI_g and PHI_p?
That was a typo and will be corrected in the manuscript.

Section 2.1.3: The words consistent and inconsistent are slightly out of place. What is a "consistent spcacing"? I assume you mean that the two data sets are interpolated on the same grid, isn't it. Later: "inconsistent with respect to time"? Here I am lost. Please reformulate.
We agree and have reformulated these sentences.

> *A regular grid leads to a evenly spacing between data points, which in in turns allows for a*
> *cleaner statistical analysis. The base of the regular grid is the geometric height z.*

> *By converting the ERA5 data from a pressure grid to a grid with the geometric height z, the*
> *latter grid changes from one point in time to the next with each atmospheric state.*

L 98: set (not sets)
L 123: Insert a paragraph break before "Potential temperature".
We have corrected this.

L 141: radiosonde
We have corrected this.

L 172: is the TIL indeed a measure? I suggest to simply delete this unneccessary sentence. Start with the second sentence.
We omitted this sentence.

L 175: I would agree that the importance of the TIL is that it is a tranport barrier. But what has it to do with any kind of diagostics? Please reformulate.
We reformulated that sentence.

> *The high stability in the TIL region represents a barrier to vertical motion (Gettelman et al.,*
> *2011) and is therefore important for understanding the composition of the air in the upper*
> *troposphere and lower stratosphere.*

L 181: diagnostics
We have corrected this.

L 190: is quite robust DUE TO small scale variations? What do you mean?
Instead of "is quite robust due to small scale variations" we meant "is quite robust due to small scale VERTICAL variations".
This was in comparison with the RHi gradient (gRHi), which is sensitive to the combination of dry and wet vertical layers while the mean relative humidity wRHi is not. However, we have omitted this sentence as it is misleading.

L 194: delete "measured"
We have corrected this.

L 229: criterion
We have corrected this.

L 346: season is
We have corrected this.

L 355: the word "perfomed" does not fit here
We rewrote this sentence, also due to remarks of reviewer #2

> By design the central USA location was introduced to include a region of frequent deep convection in the extratropics. There, significantly higher CAPE values are present in the summer months compared to the other regions (Taszarek et al., 2021), indicating greater deep convection activity. This leads to a stronger seasonal cycle in the vertical profiles in this region.

L 401: Like above: avoid the word "unstable" when speaking about the stratosphere
We changed it to "less stable"

L 415: its
We have corrected this.

**Reviewer #2**

This is a good paper. It addresses the formation mechanism of the extratropical tropopause inversion layer (TIL) and the different forcing mechanisms of the TIL discussed in the literature (Randel et al., 2007; Wirth and Szabo, 2007). Here baroclinic waves and radiative ($H_2O$) processes are relevant, but the time scales involved should not be ignored (and they might be different for the two mechanisms). Does the analysis provided here allow statements about radiat-ive/dynamical time scales?

The paper is based on high-resolution radiosonde data from Idar-Oberstein (but give location (lat/lon) at first mention, and the relevant time period) with ERA5 data. This is good. However, I would emphasise differences as well a similarities between ERA5 and the sondes (see below).

The paper then moves on to investigate the influence of relative humidity with respect to ice (i.e. $H_2O$) on the extratropical TIL. Further, based on ERA5, lon- gitudinal and seasonal variability of the TIL is discussed. But the paper should be quantitative and more accurate (and less vague) than saying something like ". . . reveal consistent relationships in various extratropical regions of the Northern Hemisphere under different meteorological conditions".
I am not sure if the authors agree with the assessment in this review – they do not need to do so. But a much clearer message of the paper would be very helpful. This is not clear from the present draft – in particular not in the title and the abstract. I think the paper would be more valuable if the message would be much clearer in a revised version.
Finally, while I am mentioning papers here that might be potentially of interest (and some already cited), I am certainly not suggesting the citation of particular papers.
Overall, I think this is a good helpful paper of interest to the readership of ACP. I suggest a major restructuring to make the key points of the paper clearer and more accurate in a revised version.

Comments in detail

 Abstract and title

There are guidelines for ACP papers, in particular the title, abstract, and conclud- ing section: https://www.atmospheric-chemistry-and-physics.net/policies/guidelines_ for_authors.html
Titles should be concise and consistent with the content and purpose of the article. For research articles, ACP prefers titles that highlight the scientific results/findings or implications of the study. Abstracts should have fewer than 250 words – I think the paper can be improved in this respect.
We changed the title of the manuscript to
> Relative humidity over ice as a key variable for Northern hemisphere extratropical tropopause inversion layers

We also rewrote the abstract to highlight our findings.

The paper should be very clear what the main findings are and what the advance of knowledge of the study is.

Comparison of ERA5 and a radiosonde station
As I understand the paper, the basis of the paper is a comparison between the TIL in ERA5 and in the data from a radiosonde station. After a 'validation' of the ERA5 data with the radiosonde data, further conclusions for the TIL in the latitude range of the station are drawn.

This is good, but the paper is not very clear about this. The stations is called "Idar- Oberstein", sometimes only "Germany" is mentioned, the period of radiosonde data is often not mentioned, on

other occasions the latitude/longitude of the station is mentioned – all the information is in the paper, but the reader should not be forced to search the entire paper to find the necessary information.

Most importantly, as I read the paper the results are relevant for northern hemi- sphere mid-latitudes (close to 50◦N) – is this correct?. If the authors agree, this fact should be evident in the paper, in particular abstract and title.

We reformulated the text in the manuscript to make all these points clearer.

Tropopause

The entire concept of the TIL is based on using tropopause relative coordinates. Yes, this is reported on page 7 of the manuscript, but I suggest making this concept clear upfront. Further, determination of the tropopause is not straightforward (e.g., in ERA5), there is a an extensive discussion in a recent publication (Hoffmann and Spang, 2022); this publication also addresses the issue of a fixed pressure grid and different interpolations, which might be helpful here.

Moreover, on page 6 of the manuscript, the classic WMO definition of the lapse rate tropopause is cited, however this definition leaves room for interpretation (Maddox and Mullendore, 2018). Exactly which definition of the lapse rate tro- popause has been used? As stated in the paper the classic WMO definition is old and does not necessarily take into account the use of more recent gridded and high resolution data (see e.g., Reichler et al., 2003).

The authors mention the review by Gettelman et al. (2011), which is good. How- ever, there are also other reviews of the determination of the tropopause (Hoinka, 1997) and there is also a tropopause definition based on isentropic potential vorti- city gradients (Kunz et al., 2011).

This comment triggered the addition of a longer section on tropopause definitions in the (new) section 2.3.1. For the sake of readability, we will not show this section in this document. The reader is referred to the manuscript.

Figs. 1, 2 and 3

First, I find these figures very helpful, they explain the concepts used here before more general statements are made.

However, I suggest that the scheme in Fig. 3 is closer to reality (Fig. 2); the static stability is not constant wit altitude (above 11.25 km) and the kink at 11.25 km in Fig. 3 is not seen in the real data (Fig. 2).

Actually, the scheme in figure 3 is quite realistic, the features mentioned in the scheme (minima and maximum in N^2) can be seen in the real-world example as represented in figure 2. We adjusted the scheme a little bit according to the realistic data, but the main features remain.

Further, I like Fig. 2 a lot, but I cannot see why an interpolation to a fixed altitude grid is necessary to produce the figure – doesn't the interpolation introduce an unnecessary smoothing? Most importantly, likely, the difference between sonde and ERA5 that I see in Fig. 2, is an important result. If I were an author, I would flag this result more strongly and more quantitatively in the paper (e.g. abstract, conclusions).

Since the interpolation is more or less a projection on a finer grid, smoothing is not an issue. We added some text to the description of the figure. However, the detailed description of the differences between RS and ERA5 is given in section 3.1.1.

> Note, that the real profile of high resolution data (radiosonde and ERA5) as represented in Figure 1 includes all the features of the scheme shown in Figure 3 (UT minimum of $N^2$, maximum of $N^2$, LS minimum of $N^2$). After averaging over many tropopause-centered profiles, some features might be lost in the mean profiles (e.g. Figure 2), although they are still visible in the single profiles – otherwise the profiles would be discharged in the analysis.

Equation 8

In Equation 8, a measure is defined for the deviation between ERA5 and the sondes. However, this definition is not unique. It is a choice, isn't it? The problem I see is that deviations between E and R could cancel out when integrated over a certain altitude range. That is locally there could be a substantial deviation between E and R, but $\bar{D}$ could be rather small, depending on how the range $z' - z_0$ was chosen. Why are no absolute values considered of the deviation between E and R?
We have replaced the difference plots with plots of the average absolute difference based on the new equation 8.

$$\overline{D}_{\mathrm{abs}}(\chi) := \frac{1}{z' - z_0} \int_{z_0}^{z'} |E(\chi) - R(\chi)| \mathrm{d}z \approx \frac{1}{z' - z_0} \sum_{z_0}^{z'} |E(\chi(z)) - R(\chi(z))| \Delta z \tag{8}$$

This affects the distributions of the deviations, i.e. Figures 4,5 and 6, and the interpretation of the results. The text was partly rewritten. We also added following discussion to the manuscript:

*We chose a metric including absolute values of differences in order to avoid undesired cancellation effects of positive and negative contributions. In this sense, we used a metric inspired by the L1-norm. Thus, the resulting distributions are expected to be skew and might have (exponential) decaying tails. The resulting data set is visualized in a probability bar chart and the corresponding median, mean and standard deviation for the different variables are presented in figures 4, 5, and 6, respectively.*

**Minor issues**

•       l. 10: I would not use the term "strong agreement" when two temperature profiles (say) are very similar.
See next issue below.

•       l. 10: "geographical": what is meant here is the longitudinal variation. Correct?
We rewrote this sentence to:
> *The shown correlation between radiosondes and ERA5 enables us to use ERA5 for seasonal analyses on different longitudinal regions in the northern extratropics.*

•       l. 18: "distinct and intriguing feature known as the tropopause inversion layer": here it would be helpful to report particular features of the TIL, rather than saying "intriguing". What are the most relevant properties of the TIL? Such information comes later in the introduction, but it would be good to have this up front.
We rewrote this sentence to:
> *At its upper boundary to the adjacent stratosphere, the troposphere encounters a strong temperature inversion known as the tropopause inversion layer (TIL)*

•       l. 25: why "inert substances" ? Isn't the TIL a barrier for vertical transport even if substances are not chemically inert?
That is correct. We omitted the word "inert" in the manuscript. We also rewrote the sentence, see comment for L 175.

•       "hypothesis" should be plural
Yes, we have corrected it to *hypotheses.*

- l. 51: give latitude and longitude of Idar-Oberstein

We added lat and lon here. More location details are given in Sec. 2.1.1 as well.

- l. 52: radiosonde data (not "sondes")

We have corrected this.

- l. 69: focuses

We have corrected this.

- l. 72: these thresholds look somewhat arbitrary. Are there citations? Are there any indications in the household data? Would a temperature of (say) 450 K be okay?

We rewrote this part to

> 419 profiles are discarded, 311 due to insufficient maximum height, 19 due to missing data and 89 due to unreliable values of temperature and relative humidity.

Actually, we leave the exact thresholds out, since the nonphysical values are such high that they just exceed all relevant values. Thus, it is not really important which exact value is used.

- l. 74: units should not be in italics

We use now the latex package siunits for all units throughout the manuscript.

- l. 80 latest −−→ most recent

We have changed it in the manuscript.

- l.83: state explicitly how many levels were used. State the top altitude used. Also the approximate vertical resolution in ERA5 here would be useful to report.

We used the version of ERA5 with 137 model levels, which has an approximate vertical resolution of 300 m in the tropopause region. A more detailed description of the individual model levels can be found here: https://confluence.ecmwf.int/display/UDOC/L137+model+level+definitions

We added the information about the vertical resolution to the manuscript.

> In the tropopause region, the vertical resolution is about 300 m.

- l. 85: "closest grid point" – this is always the same point in the ERA5 grid – correct? This point could explicitly be mentioned.

We clarified this in the manuscript

> For the comparison with the radiosonde data we obtained pseudo-radiosonde profiles, i.e. a vertical column at a fixed grid point. The vertical profile is extracted at the grid point 49.75° N and 7.25° E, which is the closest grid point of ERA5 to the actual location of Idar-Oberstein. (49.69° N and 7.33° E)

- l. 90: here and elsewhere "the data sets" is used, but is must be made clear that ERA5 and radiosonde is meant. It is likely better to err on the safe side and explicitly state what is meant.

We have made it clearer throughout the manuscript.

- l. 91: the "improved statistical analysis" is not obvious from the paper.

We changed the word improved to *straightforward*.

- l. 92: "height" be more precise here, geometric altitude, pressure altitude, geopotential altitude etc., is not the same thing and not available in each data set.

We mean "the geometric altitude z and corrected this in the manuscript.

•       l. 94: citation for the buoyancy speed?
These values were taken directly from the radiosonde data set (which we clarified in the text). However, we included the reference of Xu et al., 2023, where it is pointed out that high-resolution radiosondes, which are used e.g. by the Deutsche Wetterdienst DWD, have a sampling period of approximately 1-2s and a vertical resolution of about 5-10 m, which converts to a speed of approximately 5 m s$^{-1}$.

*Radiosondes use the buoyancy force to ascend, thus the vertical speed and consequently the vertical resolution is not constant. The buoyancy speed of the used radiosondes is ranging from 2 m s$^{-1}$ to 8 m s$^{-1}$ (with a mean around 5 m s$^{-1}$), returning a vertical resolution of 4 to 16m, respectively, what is usually recognized as high-resolution data (Xu et al., 2023).*

•       l. 96: what is the argument for cubic spline?
We have chosen a cubic spline so that the resulting interpolated radiosonde ascent is continuous with respect to the first derivative of the geometric altitude z. This is not guaranteed with linear interpolation. Also, splines with higher polynomials (>x$^3$) require higher computing power and offer no advantages over cubic splines in the context of this work.

*The interpolation is performed with a cubic spline, which offers sufficient accuracy for this study.*

•       l. 99: "inconsistent with respect to time" is unclear.
What was meant here was that the ERA5 model level is not at a constant geometric height but is dependent on the meteorological conditions at that time. We rewrote the sentence.

*By converting the ERA5 data from a pressure grid to a grid with the geometric height z the latter grid changes from one point in time to the next with each atmospheric state.*

•       l. 99: Which "data set"?
We meant ERA5, which is now clarified in the manuscript.

•       l 110: citation for this statement?
Figure 2 of Reutter et al. (2020) shows the vertical profiles of temperature in the upper troposphere and lower stratosphere for the extratropical North Atlantic region, which is similar to the regions of interest in this manuscript. All data points in the reference are below 240 K. We added the reference in the manuscript.

•       l. 117: quantify "slightly"
The deviation is less than 0.2% in the temperature regime of this manuscript according to Figure 4 of Murphy & Koop 2005. We added this information in the manuscript.

•       Eq. (2): I am not familiar with this approximation; is there a citation? Or an explanation of Eq. (2)? How accurate is Eq. (2)?
The approximation is based on the approximation of the ratio of the molar masses by $\varepsilon \approx 0.622$.  One can express $q$ by $q \approx \varepsilon \frac{p_v}{p}$ with $p_v$ the water vapour pressure and $p$ the ambient pressure. This allows to calculate the relative humidity w.r.t ice with the specific humidity

• Eq. (4) is stated here that as an approximation for dry air: this aspect should be made clear here. Make clear what the issue is of wet vs. non wet conditions.
We added some text and a citation about the treatment of moisture for static stability.

*This approximation is working under the assumption of dry air and returns on average too high values for the Brunt-Väisälä-frequency; moisture leads to strong decrease in the static stability, even if no phase change is triggered (Durran and Klemp, 1982). However, a moist, and commonly accepted, analog to dry static stability is still missing, although there are some attempts for a consistent treatment (Peters et al., 2022). Therefore we use dry static stability to ensure comparable results with literature which use the dry approximation (e.g. Gettelman and Wang, 2015; Birner et al., 2002; Birner, 2006; Erler and Wirth, 2011).*

• Eq. (5): I suggest to make clear (here and elsewhere) what z is – is it geo- potential height in Eq. (5)?
z is the geometric altitude. We made it clear in the manuscript as well.

• l. 155: see also other work (e.g., Reichler et al., 2003; Maddox and Mullendore, 2018) on using the classic WMO tropopause definition for modern, gridded data.
See above, we added an extra section (2.3.1) on the discussion on tropopause definitions. For the sake of readability, we will not show this section in this document. The reader is referred to the manuscript.

• l. 157: be clear about which data sets, "both" is a bit vague
We have made it clear in the manuscript (see also l. 90 and l.99 remark)

• l. 169: which period of the radiosonde data?
The exact used time frame spans from the 1st of January 2011 to the 31th of December 2019 as it is given in Section 2.1

• Fig. 2: This figure is good. I suggest adding some discussion in how-far the sondes and ERA5 do not agree.
We added a short description and a teaser to the more detailed investigation later in the manuscript.

*However, Figure 2 also shows that the results differ between radiosonde and reanalysis data. These differences are mainly based on the vertical resolution of the data sets, which is significantly lower in the case of the ERA5 data than in the radiosonde data. As a result, sharp gradients cannot be resolved as good, as can be seen in particular when looking at $N^2$.*
*A detailed comparison between the radiosonde and ERA5 data can be found in Section 3.1.1.*

• l. 172: "Another"? this paper is on the TIL.
You are completely right. We omitted this sentence.

• l. 175: Suggest stating the importance of the TIL earlier in the paper (intro- duction).
We moved this part to the introduction and rewrote the sentence (which was also mentioned in the comment for l. 25).

*Thus, a sharp TIL constitutes a strong transport barrier for trace gases and other parameters like vertical motion. Beyond this the importance of the TIL also lies within the diagnostics of upper troposphere and lower stratosphere (UTLS) structures.*

• l. 177: How is the value of 3 km chosen?
Schmidt et al. (2010) have found maxima of the TIL up to more than 2 km above the tropopause. Due to the large number of profiles from different regions and seasons that we consider from measurements and ERA5, we have decided to set the limit at 3 km above the tropopause.

- l. 182: say which features.

We rewrote this part.

*If one of the features such as UT-$N^2_{min}$ or LS-$N^2_{min}$ could not be determined, these profiles were excluded from consideration in this study. This affected 126 profiles, so that in the end 9678 profiles were included in the analysis in this study.*

- Eq. 7: p500z is an altitude (not a pressure) – the symbol is confusing.

We have changed this variable from $p500_z$ to $z_{p500}$ to make clear that it is an altitude.

- sec. 3.1.1: be clear what is compared with what.

By introducing the new measure of average absolute difference, we also rewrote large parts of this section with your advice in mind.

- l. 211: stating (e.g.) "upper troposphere" is not enough here, the exact range z' – z0 that was used should be reported.

We provided this information in the manuscript.

- l. 220: are $--\rightarrow$ is

We have corrected this.

- l. 223: quantify "thin"

We omitted the word "thin" and added numbers in the manuscript

*(approximately between z = 0 m to z = 500 m)*

- l. 226: do you really want to give three significant figures here?

We reduced the precision of the given values to two decimal places.

- l. 241: is this statement consistent what is shown in Fig. 2?

Since we rewrote the discussion on the differences between the data sets, we omitted this sentence

- l. 244: "averaged" over which region?

*Averaged relative humidity with respect to ice* is the description of the variable wRHi (Eq. 7) averaged accordingly from the geometric height of the 500hPa surface to the TIL height. We added a reference to Eq. 7 to avoid confusion.

- l. 269: I agree, but Fig. 2 also shows the limitations.

Yes, the limitations are mentioned during the description of figure 2.

- l. 272: which "data"?

We have added the information that this is radiosonde and ERA5 data in the text.

- l. 275: Unclear sentence, two times wRHi?

We rewrote the sentence

- l. 275: give the Fig. where the PDF can be seen

We rewrote that sentence.

*We find in Fig. 8 that the probability density function (PDF) of the TIL strength is correlated with higher values of averaged relative humidity with respect to ice (wRHi).*

- l. 286: report the latitude range for which the comparison is valid.

We added the information.

> [...]thus, the next step is to investigate three additional regions representative for northern hemisphere extratropics at a similar latitude with different meteorological conditions but at different longitudes.

- l. 289: resolution or vertical resolution?

We mean the **vertical** resolution and clarified this in the text.

- l. 291: again, point to the figure in question here, merge wit the next sen- tence (we show). . .

We rewrote this part

> In addition to the strength of the TIL, we show in Fig. 9 distributions of the TIL depths dTIL distributed into different classes of averaged relative humidity with respect to ice wRHi. Here, the depth of the TIL also shows a correlation with RHi which is discussed in this section.

- l. 293: "the TIL depth dTIL is shifted to lower values" – this is clear from Fig.9. But then (l. 294) "the depth of the TIL is always in the range" – so does DTIL change with humidity or not? Where would I see the "second mode"? In 9b? More help for the reader? I find this discussion somewhat confusing here.

We added some text for clarification and tried to make our arguments clear.

- l. 296: it is not clear why there is an "artefact" here and what this implies.

The form and existence of the double modes depend on the limit values of the categories. Without classification, there are no double modes. If one shifts the limit values slightly, then modes appear and disappear. With more categories, the double modes can be eliminated, but we wanted to be consistent with sTIL categories. That is why we decided to use the criterion of three categories with 33% of the data points each for the thresholds.

We rewrote this part to:

> This formation of a double mode is an artefact of the categorization process, in which the aim was to ensure that one third of the data per category was included. These double modes are therefore not of a physical nature.

- Fig. 9; mention a and b in the caption.

We added the information.

- l. 302: "sharpening the TIL . . . depth" – do you have an argument why?

We added some text, however, the interpretation remains difficult.

- l. 306: state here immediately which three regions. I think the main point is longitudinal variation here.

We add the longitudinal variation.

- l. 328: 328: "to act" $-\!\!-\!\!\rightarrow$ "acts"

We have corrected this.

- l. 337: give section/figures for "previous findings"

we added the reference.

- l. 340: quantify the "differences" found here

We included following part in the manuscript

> Within the first 3 km above the tropopause the static stability is on average 0.48 *10^-4 s^-1 in spring compared to autumn and 0.22 *10^-4 s^-1 higher in winter compared to summer

- l. 348: is this true? I do not see the strong increase in RHi in Fig. 2

You are right, we rewrote this sentence.

*At the same time, there is a strong increase in RHi above the first km above the tropopause.*

- l. 354: "could be due" sounds rather speculative.

We agree. We added a reference, to support this and reformulated that part in the manuscript.

*By design the central USA location was introduced to include a region of frequent deep convection in the extratropics. There, significantly higher CAPE values are present in the summer months compared to the other regions (Taszarek et al., 2021), indicating greater deep convection activity. This leads to a stronger seasonal cycle in the vertical profiles in this region.*

- l. 366: do both "effects" have similar time scales? Would this not be important?

We added some text. However, a timescale analysis is beyond the scope of the study.

*The evolution of baroclinic instabilities takes place within days, whereas radiative processes might act on time scales up to few days (if the water vapor concentration is not changed drastically in between). However, cloud processes might act on much shorter scales (minutes to hours). Thus, a clear attribution remains difficult or even impossible.*

- l. 373: compared

We have corrected this.

- l. 375: confused about "polar" and "summer" in this sentence

We refer here to Fig. 6 of Randel et al. (2007) stating that "Over polar regions, the maximum in N2 above the tropopause is substantially stronger in the summer hemisphere."

- l. 378: maximum in what?

We made it clear in the manuscript.

*The region over the Northern Pacific (Fig. 12c) shows a maximum of sTIL values in the winter and spring months, and a similar minimum in autumn and summer.*

- l. 381: what are the time scales in question here? Should not be forgotten.

See comment above (l.366)

- l. 384: amplitude yes, but what about time scales?

See comment above (l.366)

- l386: really "upper troposphere"? That means below the tropopause?

The "upper troposphere" is related to the mean moisture, which is expressed with wRHi. We clarified this in the manuscript.

*Altogether, consideration of seasonal and geographic differences reveals the overall robust relationship between the strength of TIL and average humidity with respect to ice in the upper troposphere (wRHi).*

- l. 387: "different" sounds vague here.

The differences are discussed in Section 3.3

- l. 391: give latitude and longitude of Idar-Oberstein, give length of the time period

We added this information in the manuscript.

*The example of the long-term series of radiosonde measurements at the Idar-Oberstein station (Germany, 49.69° N, 7.33° E) of the German Weather Service clearly showed that the strength of the TIL is strongly related to the relative humidity in the upper troposphere.*

- l. 393: same location is not clear; ERA5 does not have a grid point at Idar-Oberstein.

We use the grid point at 49.75N, 7.25E. We clarified this in the manuscript.

*This can be seen not only in the spatially high-resolution measurement data, but also in the reanalysis data from ERA5 at approximately the same location (ERA5 grid point at 49.75° N, 7.25° E).*

- l. 398: this is not important, but meteorologists typically talk about high temperatures, not warm temperatures.

We corrected this in the manuscript.

- l. 401: at which altitude?

We added the information in the manuscript.

*Regarding the relative humidity, one can see that the model data is slightly moister (approx. 1.2 % RHi) compared to the radiosondes in the upper tropopsphere as well as in the lower stratosphere*

- l. 401: "too stable": compared to what?

We added the information in the manuscript.

*The static stability profiles show a too stable behaviour in the UT and a too unstable behaviour in the LS of ERA5 compared to the radiosondes.*

- l. 418: provide the code on a web-page, not only "upon request ".

The code is now provided here: https://zenodo.org/records/10604349. We added this information in the manuscript.

- l. 420: It would be good to report not only the raw data, but to also create a location where the processed data of this study are available (say TIL strength).

We provided the raw data together with the used software, but we will not provide the processed data.

- l. 427: give correct "spelling" of ECMWF

We corrected *forecasting* to *forecasts*.

- l. 479: correct authors list?

We included the missing name.

- l. 490: give page rage for the citation.

We added the page range in the references.

**References**

Schmidt, T., J.-P. Cammas, H. G. J. Smit, S. Heise, J. Wickert, und A. Haser. „Observational Characteristics of the Tropopause Inversion Layer Derived from CHAMP/GRACE Radio Occultations and MOZAIC Aircraft Data". Journal of Geophysical Research: Atmospheres 115, Nr. D24 (27. Dezember 2010): 2010JD014284. https://doi.org/10.1029/2010JD014284.

Taszarek, M., Allen, J. T., Marchio, M., and Brooks, H. E.: Global climatology and trends in convective environments from ERA5 and rawinsonde data, npj Climate and Atmospheric Science, 4, https://doi.org/10.1038/s41612-021-00190-x, 2021

Xu, H., Guo, J., Tong, B., Zhang, Jinqiang, Chen, T., Guo, X., Zhang, Jian, Chen, W., 2023. Characterizing the near-global cloud vertical structures over land using high-resolution radiosonde measurements. Atmos. Chem. Phys. 23, 15011–15038. https://doi.org/10.5194/acp-23-15011-2023

---

## Referee Report (RR1)

*Second review of*

**"The extratropical tropopause inversion layer and its correlation with relative humidity (original title) "**

*by Daniel Köhler et al.*

**1   General**

This constitutes the second review of the paper by Köhler et al., submitted to ACP. I see that the manuscript has seen a lot of work; I like in particular the choice of the new metric (new Eq. 8). In spite of the work invested, I suggest a thorough proof reading of the manuscript to remove some smaller grammatical errors (some examples are given below).

As I said in the first review, "I think this is a good helpful paper of interest to the readership of ACP". And "I think the paper would be more valuable if the message would be much clearer in a revised version". I think that some further work is needed on the manuscript to make it clearer and more accurate in a revised version.

**2   Comments in detail**

**2.1   Abstract and title**

In my first review I mentioned the guidelines for ACP papers, in particular the title, abstract, and concluding section:

https://www.atmospheric-chemistry-and-physics.net/policies/guidelines_for_authors.html

The title has been changed and improved.

However, I am afraid that the abstract is still longer than 250 words (see ACP recommendations). I suggest addressing this issue. I think abstract of the paper could also be a bit clearer about the findings of the paper; would it not be of interest to the reader what the "consistent relationships" and the "differences" are?

I would also suggest replacing "extratropical" with "mid-latitude" in the title, as this paper is not addressing polar issues.

**2.2 Comparison of ERA5 and a radiosonde station**

In my first review I stated: "As I understand the paper, the basis of the paper is a comparison between the TIL in ERA5 and in the data from a radiosonde station. After a 'validation' of the ERA5 data with the radiosonde data, further conclusions for the TIL in the latitude range of the station are drawn."

The paper has certainly improved as the information on "Idar-Oberstein, Germany" is now much clearer.

However, as stated earlier, the results are relevant for northern hemisphere mid-latitudes (close to $50°N$) – is this correct?. I suggest to be clearer about this. The title has changed to "extratropics" (which is not the same thing as mid-latitudes) but this point is not reflected (e.g.) in the abstract. I think the paper should be very clear (in particular in the abstract and in the conclusions) about the range of validity of the analysis.

If the authors argue that the analysis focuses on Northern hemisphere mid-latitudes (e.g., ls. 550/551), my suggestion is to replace "extratropical"' with "midlatitude".

**2.3 Timescales**

In the first review it was stated: "Here baroclinic waves and radiative ($H_2O$) processes are relevant, but the time scales involved should not be ignored (and they might be different for the two mechanisms). Does the analysis provided here allow statements about radiative/dynamical time scales?"

I do not think that this question is well answered (manuscript and reply); perhaps, statements about time scales cannot be made based on the analysis, however this point could come across clearer in the paper.

**2.4 Tropopause**

As stated in my first review, "the entire concept of the TIL is based on using tropopause relative coordinates".

The paper now cites a number of reviews (e.g., Reichler et al., 2003; Maddox and Mullendore, 2018, which is good), however, Maddox and Mullendore (2018)

report two different interpretations (implementations) of the classic WMO tropopause definition (see their appendix); it is still not clear from the paper which interpretation was used here.

**2.5 Radiosondes**

If I understand the paper correctly, radiosonde data from one (DWD) station at Idar-Oberstein are considered here. This, to me, seems to be the "anchor"-point of the ERA5 analysis (see also section 2.2).

However, when the radiosonde/ERA5 comparison is described (p. 13, Figs. 4-6), only "the radiosondes" are mentioned. But which sondes, which station, which time period? My guess is that Idar-Oberstein sondes were used here (see also data availability statement), but this point should be clear from the paper. I also suggest adding this information (which radiosondes?) to the captions of Figs. 4, 5, and 6.

Is it also true that the Idar-Oberstein sondes were assimilated in ERA5? I think the answer is yes. This information could be added to the paper, e.g., in line 328.

**2.6 Findings of Kunkel et al. (2016)**

The study aims at corroborating the findings of Kunkel et al. (2016); e.g., l. 81. Kunkel et al. (2016) state for example "Furthermore, updrafts moisten the upper troposphere and as such increase the radiative effect from water vapor" – this seems to emphasise the radiative effects on the TIL – this emphasis is not obvious from the present manuscript. Perhaps a better link to the paper (and results) by Kunkel et al. (2016) could be made.

**3 Minor Points**

- l. 6: which radiosonde data?

- l. 13: extent $\longrightarrow$ extend

- l. 14: "relationships" between which variables?

- l. 15: "Differences" between what?

- l. 134: using an adverb ("evenly"') is not appropriate here.

- l. 202: attempts $\longrightarrow$ ways

- l. 212: the point is the "vertical gradient" – correct?

- l. 303: suggest replacing "extratropics'" with "midlatitudes".

- l. 321: exponential $\longrightarrow$ exponentially

- l. 378: weakening $\longrightarrow$ weaken

- l. 482: I would formulate: "mid-latitude region"

- l. 539: What is casing the variable relative humidity?

- l. 540: "sharpening of the TIL"?

- l. 540: drop "the" after "for"

- l. 548: results

- l. 554: drop "we can state that"

- l. 554: "data are"

- l. 554: "kind of investigation"

- l. 555: here it would be helpful to state exactly which findings of Kunkel at al. (2016) are corroborated.

- Fig. 4: state in the caption which altitude range is considered to be UT and LS.

- References: journals should not be in upper-case letters (e.g. l. 579).

**References**

Maddox, E. M. and Mullendore, G. L.: Determination of Best Tropopause Definition for Convective Transport Studies, J. Atmos. Sci., pp. 3433–3446, URL https://doi.org/10.1175/JAS-D-18-0032.1, 2018.

Reichler, T., Dameris, M., and Sausen, R.: Determining the tropopause height from gridded data, Geophys. Res. Lett., 30, 2042, https://doi.org/10.1029/2003GL018240, 2003.

---

## Author Response (AR2)

**Reply to Reviewer**

We thank both reviewers for their work and are pleased that they recognize the work we have put into reformulating the manuscript. In the following, we will address the points that arose during the second review.

*Questions and remarks of the reviewers are marked in orange,* reply of the authors are marked in black and *changes to the manuscript are marked in blue.*

**Report #1**

*To my view, the authors did an excellent job to revise their previous version and the paper is much better now. Only a few typos should be corrected before printing.*

*Line 23: Check whether troposphere or rather tropopause is the more appropriate word here.*
We reworte the sentence to
> *About 20 years ago, Birner et al. (2002) were able to demonstrate with high-resolution radiosonde data that sometimes the upper troposphere and lower stratosphere (UTLS) region encounters a strong temperature inversion known as the tropopause inversion layer (TIL).*

*L 29: lower atmosphere is not clear enough. I prefer troposphere.*
We corrected this.

*Check the sentence in Ll 30,31. It is unclear.*
We omitted that sentence

*L 43: irreversibly*
We corrected this.

*L 162: constituteS, and later serveS*
We corrected this.

*L 258: replace "live"*
We rewrote the sentence to
> *This means that all profiles are now in the same coordinate system and can be averaged to obtain mean profiles of temperature, humidity and static stability.*

*L 304: I suggest to write "a key quantity"*
We applied the suggestion.

*L 396: formation*
We corrected this.

*L 475: Please try to reformulate. It is a bit ugly that the word "average" appears twice.*
We rewrote the sentence to
> *The averaged RHi for central Europe show only small seasonal differences in the troposphere.*

L 507: regionS
We corrected this.

**Report #2**

**1 General**

This constitutes the second review of the paper by Köhler et al., submitted to ACP. I see that the manuscript has seen a lot of work; I like in particular the choice of the new metric (new Eq. 8). In spite of the work invested, I suggest a thorough proof reading of the manuscript to remove some smaller grammatical errors (some examples are given below).

As I said in the first review, "I think this is a good helpful paper of interest to the readership of ACP". And "I think the paper would be more valuable if the message would be much clearer in a revised version". I think that some further work is needed on the manuscript to make it clearer and more accurate in a revised version.

**2 Comments in detail**

**2.1 Abstract and title**

In my first review I mentioned the guidelines for ACP papers, in particular the title, abstract, and concluding section:

https://www.atmospheric-chemistry-and-physics.net/policies/guidelines_for_authors.html

The title has been changed and improved.

However, I am afraid that the abstract is still longer than 250 words (see ACP recommendations). I suggest addressing this issue. I think abstract of the paper could also be a bit clearer about the findings of the paper; would it not be of interest to the reader what the "consistent relationships" and the "differences" are?

I would also suggest replacing "extratropical" with "mid-latitude" in the title, as this paper is not addressing polar issues.

We replaced „extratropical" with „midlatitude" in the title and throughout the whole text. In addition, we have shortened the abstract to exact 222 words.

**2.2 Comparison of ERA5 and a radiosonde station**

In my first review I stated: "As I understand the paper, the basis of the paper is a comparison between the TIL in ERA5 and in the data from a radiosonde station. After a 'validation' of the ERA5 data with the radiosonde data, further conclusions for the TIL in the latitude range of the station are drawn."

The paper has certainly improved as the information on "Idar-Oberstein, Ger- many" is now much clearer.

However, as stated earlier, the results are relevant for northern hemisphere mid- latitudes (close to 50°N) – is this correct?. I suggest to be clearer about this. The title has changed to "extratropics" (which is not the same thing as mid-latitudes) but this point is not reflected (e.g.) in the abstract. I think the paper should be very clear (in particular in the abstract and in the conclusions) about the range of validity of the analysis.

If the authors argue that the analysis focuses on Northern hemisphere mid-latitudes (e.g., ls. 550/551), my suggestion is to replace "extratropical"' with "midlatitude".

As we explain in the introduction and in Section 3.3, we have chosen the four locations so that they reflect different meteorological characteristics within the midlatitudes. We believe that we have already written this clearly enough.

We agree with you that the term extratropics was not precise here and, following your suggestion, we have used the term midlatitudes throughout the manuscript.

**2.3 Timescales**

In the first review it was stated: "Here baroclinic waves and radiative ($H_2O$) pro- cesses are relevant, but the time scales involved should not be ignored (and they might be different for the two mechanisms). Does the analysis provided here allow statements about radiative/dynamical time scales?"

I do not think that this question is well answered (manuscript and reply); perhaps, statements about time scales cannot be made based on the analysis, however this point could come across clearer in the paper.

We added text at several locations in the text, indicating that our investigation is based on an Eulerian evaluation, which does not allow to determine timescales of the processes involved for changing the relative humidity. Such approach would require a Lagrangian analysis, which is, however, beyond the scope of this study.

> *However, a clear distinction between the different processes or the involved timescales is not possible on the basis of RHi values only.*

*With our Eulerian approach of evaluating RHi at a given location and time, respectively, a further determination of the dominant processes and/or their timescales is not possible. For such investigations, a Lagrangian approach would be necessary.*

**2.4 Tropopause**

As stated in my first review, "the entire concept of the TIL is based on using tropopause relative coordinates".

The paper now cites a number of reviews (e.g., Reichler et al., 2003; Maddox and Mullendore, 2018, which is good), however, Maddox and Mullendore (2018)

We added this information at the end of section 2.3.1:

*In this paper we use the pure WMO criterion to determine the tropopause. For details, the reader is referred to the source code that we have provided (https://zenodo.org/records/10604349).*

**2.5 Radiosondes**

If I understand the paper correctly, radiosonde data from one (DWD) station at Idar-Oberstein are considered here. This, to me, seems to be the "anchor"-point of the ERA5 analysis (see also section 2.2).

However, when the radiosonde/ERA5 comparison is described (p. 13, Figs. 4-6), only "the radiosondes" are mentioned. But which sondes, which station, which time period? My guess is that Idar-Oberstein sondes were used here (see also data availability statement), but this point should be clear from the paper. I also suggest adding this information (which radiosondes?) to the captions of Figs. 4, 5, and 6.

We added the information about Idar-Oberstein in the captions of Fig. 4, 5, and 6.

Is it also true that the Idar-Oberstein sondes were assimilated in ERA5? I think the answer is yes. This information could be added to the paper, e.g., in line 328.

We added some text for hopefully making clearer that we just use the sondes from Idar-Oberstein. We added also the information that the used radiosondes were assimilated into the ERA5 data.

*For the first investigation, nearly 10,000 high-resolution radiosonde ascents from **one distinct** weather station in Germany (Idar-Oberstein) are analyzed.*

*This study is partly based on radiosonde data from a **single** measurement site at Idar-Oberstein (Germany, 49.69° N, 7.33° E)*

*The radiosonde data as described above (sec. 2.1.1) is also assimilated into the ERA5 data set.*

**2.6 Findings of Kunkel et al. (2016)**

The study aims at corroborating the findings of Kunkel et al. (2016); e.g., l. 81. Kunkel et al. (2016) state for example "Furthermore, updrafts moisten the upper troposphere and as such

increase the radiative effect from water vapor" – this seems to emphasise the radiative effects on the TIL – this emphasis is not obvious from the present manuscript. Perhaps a better link to the paper (and results) by Kunkel et al. (2016) could be made.

Actually, Kunkel et al. suggested that the combination of (reversible) adiabatic processes (as transport) and diabatic processes (as mixing or radiative cooling) lead to a sharpening of the TIL in combination with an enhanced relative humidity. Using our detection method (in terms of enhanced RHi values)  we found statistical evidence of the correlation although we cannot distinguish between the different processes (adiabatic vs. Diabatic). We added some text in the result section to make clear the connection to the former study by Kunkel et al.

> *This result is in agreement with the findings by Kunkel et al. (2016) for a single case analysis; adiabatic processes (transport by atmospheric flows) and diabatic processes (as mixing or radiative cooling) lead to enhanced values of RHi*

**Minor Points**

**l. 6: which radiosonde data?**
To keep the abstract short, we have omitted the location in the abstract. In our opinion, it would go too far to describe the exact location and position of the radiosondes in the abstract. However, this is described in detail in the data section.

**l. 13: extent −−→ extend**
We corrected this.

**l. 14: "relationships" between which variables?**
We rewrote the sentence to
> *These analyses reveal consistent TIL properties in various extratropical regions of the Northern Hemisphere under different meteorological conditions.*

**l. 15: "Differences" between what?**
We rewrote the sentence to
> *However, differences in the strength of the dependence of TIL properties on relative humidity over ice are evident between the different regions.*

**l. 134: using an adverb ("evenly"') is not appropriate here.**
We corrected this.

**l. 202: attempts −−→ ways**
We corrected this.

**l. 212: the point is the "vertical gradient" – correct?**
No, it is actually a point because it is the height at which the minimum of the vertical temperature profile is reached. See Highwood and Hoskins (1998), which was already given as reference in the revised version.

l. 303: suggest replacing "extratropics"' with "midlatitudes".
We corrected this.

l. 321: exponential −−→ exponentially
We corrected this.

l. 378: weakening −−→ weaken
We corrected this.

l. 482: I would formulate: "mid-latitude region"
We corrected this.

l. 539: What is causing the variable relative humidity?
We are not sure what is involved in this question. The definition of relative humidity is given in section 2.2 with equations 1 and 2.

l. 540: "sharpening of the TIL"?
We corrected this.

l. 540: drop "the" after "for"
We corrected this.

l. 548: results
We corrected this.

l. 554: drop "we can state that"
We corrected this.

l. 554: "data are"
We corrected this.

l. 554: "kind of investigation"
We corrected this.

l. 555: here it would be helpful to state exactly which findings of Kunkel at al. (2016) are corroborated.
We have rewritten the sentence to:
*The connection between relative humidity and TIL features is robust and corroborates the former findings in model studies by Kunkel et al. (2016), who stated that diabatic and adiabatic processes can have similar amplifying effects on the TIL.*

Fig. 4: state in the caption which altitude range is considered to be UT and LS.
We included this.

References: journals should not be in upper-case letters (e.g. l. 579).
We corrected this.